# Snow depth sensitivity to mean temperature, precipitation, and elevation in the Austrian and Swiss Alps

Matthew Switanek[1], Gernot Resch[1], Andreas Gobiet[2], Daniel Günther[2], Christoph Marty[3], and Wolfgang Schöner[1]

[1]Department of Geography and Regional Science, University of Graz, Graz, Austria
[2]GeoSphere Austria, Vienna, Austria
[3]WSL Institute for Snow and Avalanche Research SLF, Davos, Switzerland

**Correspondence:** Matthew Switanek (matthew.switanek@uni-graz.at)

**Abstract.** Snow depth plays an important role in the seasonal climatic and hydrological cycles of alpine regions. Previous studies have shown predominantly decreasing trends of average seasonal snow depth across the European Alps. Additionally, prior work has shown bivariate statistical relationships between average seasonal snow depth and mean air temperature or precipitation. Building upon existing research, our study uses observational records of in situ station data across Austria and
Switzerland to better quantify the sensitivity of historical changes in seasonal snow depth through a multivariate framework that depends on elevation, mean temperature and precipitation. These historical sensitivities, which are obtained over the 1901/02-1970/71 period, are then used to estimate snow depths over the more recent period 1971/72-2020/21. We find that the year-to-year estimates of snow depths, which are derived from an empirical-statistical model (SnowSens), that rely solely on the historical sensitivities are nearly as skillful as the operational SNOWGRID-CL model used by the weather service at GeoSphere
Austria. Furthermore, observed long-term changes over the last 50 years are in better agreement with SnowSens than with SNOWGRID-CL. These results indicate that historical sensitivities between snow depth, temperature and precipitation are quite robust over decadal-length scales of time, and they can be used effectively to translate expected long-term changes in temperature and precipitation to changes in seasonal snow depth.

## 1 Introduction

Snow on the ground is an important component of the hydrological cycle, the climate system, and mountain ecosystems throughout the world (Beaumet et al., 2021; Beniston et al., 2018; Gobiet et al., 2014; Notarnicola, 2022). The timing of snowfall, along with its accumulation, has profound implications on water resources (Viviroli et al., 2011; Colombo et al., 2023; Avanzi et al., 2024), mountain tourism (Elsasser and Bürki, 2002), and mountain hazards such as avalanches (Marty et al., 2017b). Understanding the impact that climate change has on this valuable resource is therefore essential in order to
better assist regional planning and preparedness.

Correctly quantifying changes in the climate system or hydrologic cycle, generally require robust measurements with sufficiently long time series of high data quality. Historically, there are two quantities that have been measured in situ by national hydrometeorological services that fulfill these criteria: 1) snow depth and 2) the depth of snowfall. Depth of snowfall is defined

as freshly fallen snow that accumulated on a snow board during a standard observing period of 24 hours, while snow depth is the total accumulated snowpack from the ground surface to the snow top (Haberkorn, 2019). In the European Alps, there is a long history of snow depth and snowfall measurements that date back to the 19th century (Scherrer et al., 2013). Over the more recent past, satellite data are becoming increasingly important to provide information on specific spatial patterns (Hüsler et al., 2014; Lievens et al., 2019). However, the measurement record of satellite data is reasonably short and unfortunately cannot provide the same quality of information as in situ observations when it comes to quantities involving snow depth (Lievens et al., 2022). Other variables describing snow characteristics, such as snow water equivalent (SWE), were introduced later in the measurement record and with a lower density network (see e.g. Haberkorn (2019)).

Matiu et al. (2021) was one of the first studies to provide an extensive and comprehensive analysis of changes in snow depth for the period 1961-2020 that truly covers the entire region of the Alps. It is worth mentioning their great effort in merging many individual stations across different institutions and networks. In their study, Matiu et al. (2021) were able to show predominantly a decreasing trend of snow depth across the Alps. In addition to the regional differences that they found in the trends, they also showed a strong elevation dependence of snow depth trends. However, this elevation dependence of the snow depth trends is conflated to some degree with the fact that stations at higher elevations also typically receive more snow. As a result, there is also a benefit in investigating whether the relative changes in snow depth are increasing or decreasing with elevation (Laternser and Schneebeli, 2003; Marty and Blanchet, 2012; Marty et al., 2023).

Beyond quantifying historical trends in snow depth and snowfall themselves (Bertoldi et al., 2023), it is additionally useful to attribute these changes to a certain set of physical drivers. The accumulation of snow depth over a season is primarily driven by temperature and precipitation (Sippel et al., 2020; Pepin et al., 2022). There have been several prior studies that have linked changes in snow depth, at different elevations, to changes in air temperature and precipitation (Scherrer and Appenzeller, 2006; Morán-Tejeda et al., 2013; Sospedra-Alfonso et al., 2015; Scalzitti et al., 2016; Schöner et al., 2019; Monteiro and Morin, 2023). Overall, these studies have shown snow depth being strongly related to air temperature at low elevations and to precipitation at high elevations (Morán-Tejeda et al., 2013; Schöner et al., 2019). However, these studies also experience to some extent a conflation between snow depth quantities and elevation. Furthermore, the statistical relationships shown are often correlations, which do not capture how much snow depth would change, for example, as a function of air temperature.

Our study aims to extend prior work in a number of ways. First, we start by computing anomalies of snow depth, temperature, and precipitation data for stations across Austria and Switzerland. This step removes regional and elevation-dependent climatological differences, thereby allowing us to better quantify anomalous or relative changes. Next, we observe how sensitive snow depth has been in the historical record to anomalous changes in temperature and precipitation for stations within specified elevation bands. Then, we use the historically derived sensitivities to construct an empirical-statistical model to estimate seasonal snow depth provided seasonal anomalies of temperature and precipitation. Lastly, we evaluate model performance. The model is calibrated over the period 1901/02-1970/71, and evaluation is performed over the period 1971/72-2020/21. Model evaluation is also compared to that of Geosphere's SNOWGRID-CL model for the Austrian domain. Our primary objective is to provide an effective yet easy to interpret method to translate expected long-term changes in temperature and precipitation to changes in seasonal snow depth.

## 2 Data

Stations with daily measured snow depth were collated from the Austrian and Swiss meteorological services GeoSphere Austria, Hydrographischer Dienst (HD), MeteoSwiss, and the WSL Institute for Snow and Avalanche Research (SLF). Some of these stations began record keeping in the 1880s, while many others became active in the early part of the 20th century. At the time that the authors collected the data, these stations primarily have data coverage through the spring of 2021. While seasonal values of snow depths, mean temperature, and precipitation reflect the accumulations or averages spanning from one year to the next, for the duration of the paper we simply use the year in which the season ends. So, for example, the year 2021 would refer to the season November 2020 - March 2021. There are a total of 291 snow stations with 107 stations in Austria and 184 stations in Switzerland. The stations range in elevation between 121 and 2536 meters with a mean height of 1097 meters (see Figure 1). While we use the non-homogenized snow depth measurements in this study, it is not expected that the results would substantially change using homogenized snow data. Through personal communication, the authors of a recent homogenization study in the Alps (i.e., Resch et al. (2022)) have indicated that there are not any systematic changes in snow depth one way or the other as a result of the homogenization procedure (Marcolini et al., 2019; Buchmann et al., 2022). Furthermore, the primary focus of this study is to present a useful methodology or approach to quantify the influence that anomalous seasonal temperature and precipitation has on snow depth.

Monthly homogenized temperature and precipitation data for Austria are obtained from GeoSphere (previously ZAMG) as part of the HISTALP data set (i.e., Historical Instrumentation Climatological Surface Time Series of the Greater Alpine Region, https://www.zamg.ac.at/histalp/dataset/station/csv.php), while homogenized precipitation and temperature data for Switzerland are obtained from MeteoSwiss (https://www.meteoschweiz.admin.ch/service-und-publikationen/applikationen/ext/climate-tables-homogenized.html). There are a total of 43 temperature stations and 48 precipitation stations in Austria (see Figure 1), while in Switzerland, there are a total of 29 temperature stations and 27 precipitation stations.

We evaluate the statistical model across Austria against a dynamical snow cover model SNOWGRID-CL (Olefs et al., 2020), which is a simplified climate version of the operational snow model SNOWGRID (Olefs et al., 2013) at GeoSphere Austria. The model was developed for climatological simulations such as long historical runs and future scenarios. It relies on an extended degree-day scheme to approximate snow ablation from air temperature and the shortwave radiation balance (see Olefs et al. (2020) for further details). The model is forced with an observation-based gridded dataset of air temperature and precipitation (SPARTACUS v2.1, Hiebl and Frei (2016, 2018)) and simulates daily fields of snowpack properties (i.e., snow depth, SWE) at a spatial resolution of 1km x 1km over the Austrian domain. The model output is updated daily and stands publicly available at Geosphere Austria's Data-Hub (https://public.hub.geosphere.at/public/datahub.html?id=snowgrid_cl-v2-1d-1km/filelisting&anonymous=true#/snow_depth/).

### 2.1 Evaluation Metrics

Performance of different modeled time series of snow depths are compared using the root mean squared error (RMSE) statistic. This metric is used becuase it measures how well the estimates from the model covary with observations, while it additionally

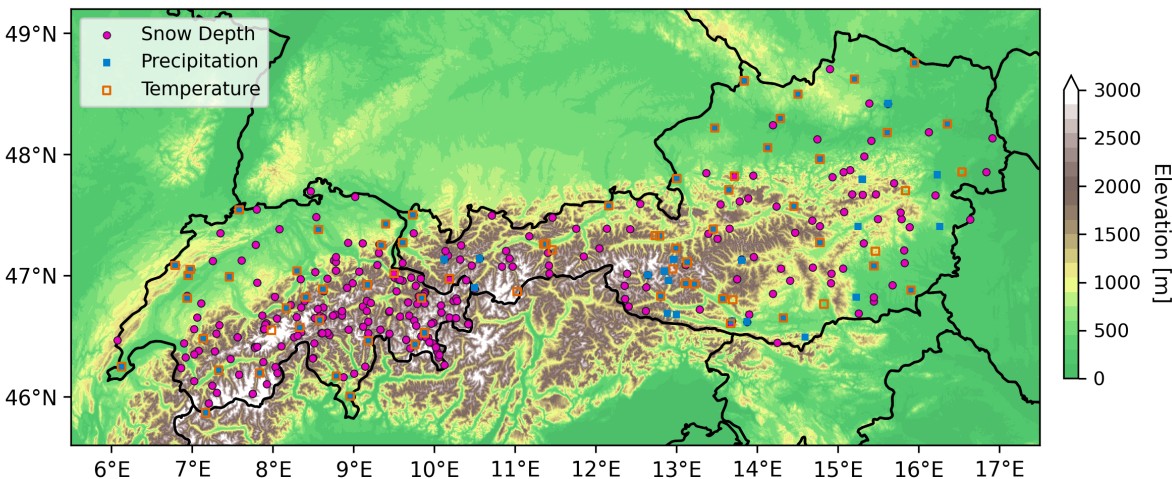

**Figure 1.** Study region with elevation. Stations with historical measurements of daily snow depth are plotted as the magenta circles. Monthly homogenized temperature and precipitation stations are plotted as the hollow orange boxes and the filled blue squares, respectively.

reflects whether or not there is any systematic mean bias between the two time series. This metric is computed using both time series of absolute (not to be confused with the mathematical absolute value, but the raw modeled and observed values expressed in centimeters) and anomalous values. The modeled RMSE is calculated as,

$$\mathbf{RMSE}_{MOD} = \sqrt{\frac{1}{n}\sum_{i=1}^{n}\left(y_{mod,i} - y_{obs,i}\right)^2} \ . \tag{1}$$

where $y_{mod,i}$ and $y_{obs,i}$ are seasonal time series of modeled and observed snow depths at station, $i$, respectively. Likewise, $\mathbf{RMSE}_{CLIM}$, is defined as,

$$\mathbf{RMSE}_{CLIM} = \sqrt{\frac{1}{n}\sum_{i=1}^{n}\left(y_{clim,i} - y_{obs,i}\right)^2} \ . \tag{2}$$

and reflects the error associated with estimates using climatological, where $y_{clim,i}$ is the reference mean climatological snow depth at station, $i$. One can think about $y_{clim,i}$ as either a single value (i.e., the climatological mean) or a time series array with all of the values being the same. The values of $y_{mod,i}$ and $y_{clim,i}$ change depending on whether we are computing the RMSE skill score using absolute values or anomalies. For the absolute values evaluated over the period 1972-2021, $y_{obs,i}$, $y_{mod,i}$ and $y_{clim,i}$ would all contain values expressed in centimeters, where $y_{clim,i}$ are the mean seasonal snow depths, computed station by station, over the 1902-1971 calibration period. When computing the skill of the model estimated anomalies, $y_{obs,i}$, $y_{mod,i}$ and $y_{clim,i}$ all contain values expressed as % of normal (e.g., 120% of normal, which is 20% above normal), where $y_{clim,i}$

is the mean seasonal snow depth, at station $i$, over the 1902-1971 calibration period (i.e., 100%). Additionally, a RMSE skill score is also used to evaluate the performance of the models in their ability to capture observed trends. In that case, a trend of 0% per decade is treated as the climatological reference. Then, the RMSE skill score, $\mathbf{SS}_{RMSE}$, is defined as,

$$\mathbf{SS}_{RMSE} = 1 - \frac{\mathbf{RMSE}_{MOD}}{\mathbf{RMSE}_{CLIM}} \quad , \tag{3}$$

where an $\mathbf{SS}_{RMSE}$ value of 1.0 would be perfectly estimating the observations, values between 0.0 and 1.0 reflect model estimates that perform better than climatology, and values below 0.0 indicate that the model is less skillful than climatology.

## 3  Methods

### 3.1  Seasonal Snow Depth

In this paper, we focus on snow depth averaged over the November-March season. Our first goal is to investigate historical empirical relationships between mean seasonal temperature, precipitation and snow depth. And second, we use these historically derived empirical relationships to estimate changes in snow depth driven by changes in mean seasonal temperature and precipitation. During warmer months, and especially with stations at lower elevations, an observable amount of precipitation will not always translate to a measured snow depth. This would result in trying to fit a predictor time series (i.e., precipitation), which does vary, with our predictand time series that does not (i.e., snow depth). We try to minimize the number of cases where there is zero measured snow depth by excluding the months of April and May from our seasonal average, since those months often contain more stations with zero recorded snow depth. That way, we can have consistency in the lengths of our seasons for both the predictors, T and P, and our predictand, SD. Figure 2a shows the percentage of data for which the 291 stations in this study measured an average monthly snow depth greater than 0.0 cm. For example, when considering all of the Januaries between 1901-2020, there were 20,263 station-months with full data coverage. Of those, 500 recorded 0.0 cm for every day throughout January at a station. The percentage of zero measured snow depth for January is then 2.5%, which is equal to 500/20,263 (this quantity is also equal to 100% minus the percentage shown in the left-most bar of Fig. 2a). We used the season November-March, because each of those five months contained less than 20% zero measured snow depth. While the November-March season is somewhat shorter than what some other studies have used (e.g., Matiu et al. (2021); Morán-Tejeda et al. (2013)), the average anomalous November-March snow depth (anomalies are represented as % of normal, see Eq. 6) varies strongly with the average anomalous snow depth over the longer November-May season (see Fig. 2b). So, if one can skillfully estimate November-March snow depth, then these will also be skillful for a longer season such as November-May. In Figure 2c, average seasonal snow depth can be seen to vary with elevation, with higher elevations generally receiving more snow.

### 3.1.1  Observed Changes in Seasonal Snow Depth

Figure 3 shows the trends of snow depth anomalies (see Eq. 6), across the Austrian and Swiss Alps, for the period of record 1902-2021 using our four elevation bands. Here, we have computed linear trends. However, we should mention a couple of

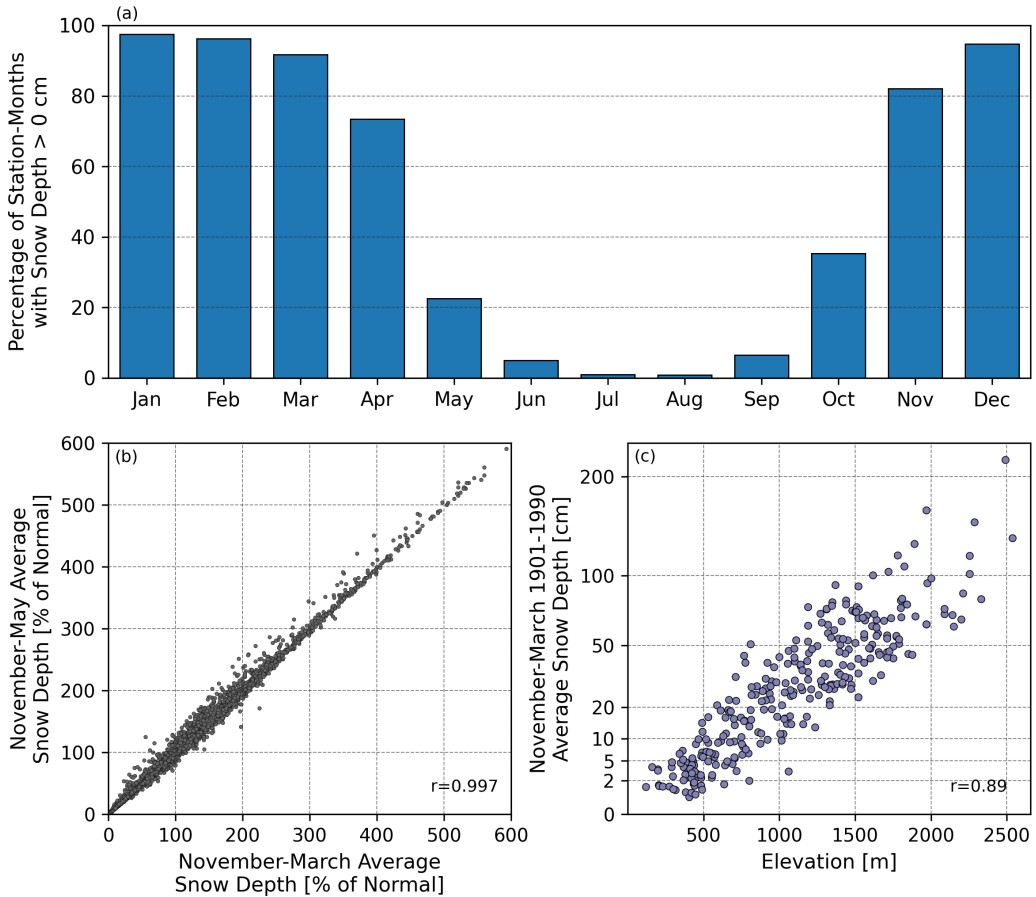

**Figure 2.** Figure 2a plots the percentage of data where the monthly average snow depth was greater than 0.0 cm. (b) plots November-March average snow depth, as percent of normal over the period 1902-2021, against the longer November-May season. (c) shows how the the average seasonal snow depth varies with elevation, with higher elevations generally receiving more snow. Note the logarithmic scaling of the y-axis in (c).

small caveats in doing this. First, the data is bounded by zero, and a negative trend line would eventually cross the origin to produce negative values of snow depth, which cannot physically happen. Second, the data does typically exhibit some level of positive skewness, and as a result the data cannot be considered truly Gaussian. In order to allow for a easily interpretable trend information, linear trends are computed over the time series. That way, even though the values are bounded by zero and also cannot be considered truly Gaussian, it can be illustrated that values have decreased over time, i.e. show a negative trend. Additionally, we primarily want to illustrate that trends have decreased across each elevation band, and that the trends have been greatest at lower elevations. For the stations below 500 meters, the decrease in snow depth is -8.1% per decade (Figure 3a), while the stations at elevations above 1500 meters exhibit less than half of the relative trend at -3.4% per decade (Figure


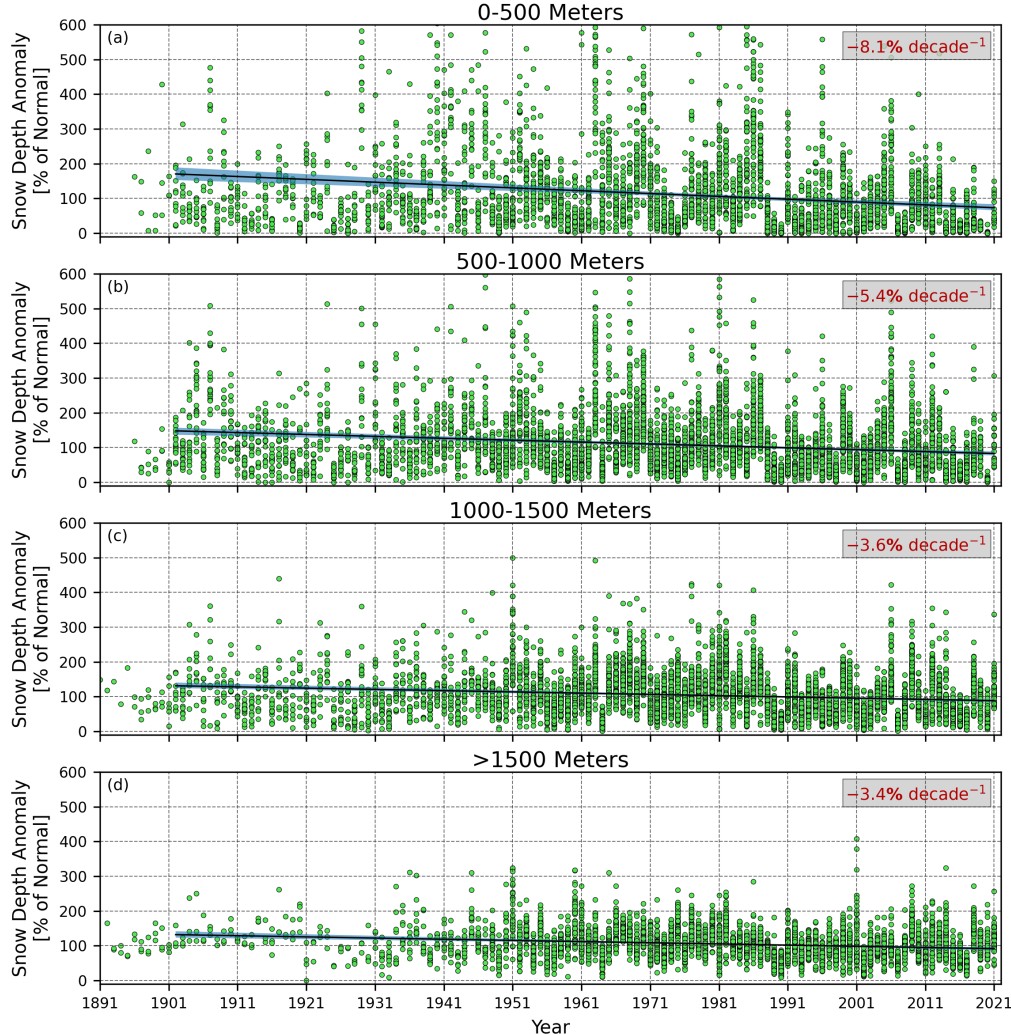

**Figure 3.** The seasonal (Nov-Mar) snow depth anomalies for all stations and all seasons are plotted as the dots for each of the four different elevation bands. Trend lines, and the relative change per decade, are also plotted. Here, we use the entire period of record. All station-season values are used to compute the historical trend. For example, the trend in Fig. 3a is the least-squares regression fit to all of the points in Figs. 3a. Bootstrapping is used to estimate 95% confidence intervals, which are shown as the blue shading.

3d). Additionally, the regime shift at the end of the 1980's described by Marty (2008) for the Alps and (Reid et al., 2016) on the global level is nicely visible, especially for the two lower elevation bands. Please refer to Appendix A to see trends over a more recent period, and the spatial patterns of the historical changes.

Different snow depth stations have different record lengths. As a result, the anomalies from the estimated mean seasonal snow depth can vary to some degree depending on which seasons are used to compute the average. In Table 1, we show a range

| | All Stations | >80% Coverage | >90% Coverage | >95% Coverage |
|---|---|---|---|---|
| 0-500m trend (1902-2021) [% decade$^{-1}$] | -8.1 | -8.1 | -7.6 | -7.8 |
| 500-1000m trend (1902-2021) [% decade$^{-1}$] | -5.5 | -5.3 | -4.9 | -4.9 |
| 1000-1500m trend (1902-2021) [% decade$^{-1}$] | -3.6 | -3.5 | -3.4 | -3.5 |
| >1500m trend (1902-2021) [% decade$^{-1}$] | -3.4 | -3.8 | -3.4 | -3.3 |
| 0-500m trend (1952-2021) [% decade$^{-1}$] | -15.7 | -15.8 | -15.3 | -15.3 |
| 500-1000m trend (1952-2021) [% decade$^{-1}$] | -10.3 | -10.3 | -10.1 | -10.3 |
| 1000-1500m trend (1952-2021) [% decade$^{-1}$] | -6.0 | -5.7 | -5.3 | -5.7 |
| >1500m trend (1952-2021) [% decade$^{-1}$] | -3.6 | -3.8 | -3.8 | -3.7 |
| 0-500m change (1992-2021 vs. 1952-1981) [%] | -46.4 | -46.7 | -45.4 | -45.6 |
| 500-1000m change (1992-2021 vs. 1952-1981) [%] | -34.6 | -34.7 | -34.6 | -35.5 |
| 1000-1500m change (1992-2021 vs. 1952-1981) [%] | -25.1 | -24.7 | -23.5 | -24.6 |
| >1500m change (1992-2021 vs. 1952-1981) [%] | -15.8 | -16.7 | -16.7 | -16.2 |
| 0-500m (number of stations) | 52 | 39 | 33 | 32 |
| 500-1000m (number of stations) | 75 | 61 | 45 | 40 |
| 1000-1500m (number of stations) | 91 | 72 | 62 | 57 |
| >1500m (number of stations) | 73 | 47 | 38 | 36 |

**Table 1.** Table values express trends and changes across different time periods for the four elevation bands. The trends are percentage changes per decade, given for the two different time periods from Fig. 3 and Fig. A1 from the Appendix, while the changes are the average percentage changes between two periods over the last 70 years. Stations have varying data lengths and coverages, which can influence mean estimation. The percentage of data coverage is evaluated over the 1951/52-2020/21 period, where 80% coverage would mean that a particular station had at least 56 years of data. The first three sets of rows show how much the trends change when using stations with more or less data coverage. The bottom set of rows gives the number of stations used to compute the trends and changes. The units of the rows are provided within the square brackets.

of historical trends depending on how much data coverage we set as a threshold. The trends are not found to change very much when only the stations with more complete data coverage are used.

## 3.2 Constructing Homogenized Temperature and Precipitation Time Series at the Snow Stations

By itself, snow depth is not a measure of how much melted water is contained in the snow mass. Many different meteorological conditions can affect the density of the snowpack or the melting of snow. As a result, we cannot directly infer how much precipitation had fallen by the snow depth measurements themselves. The precipitation could have fallen as rain, while precipitation falling as snow can accumulate at varying densities. Ultimately, we want to quantify historical changes in mean temperature

and precipitation and how these have translated into changes in snow depth. Therefore, since many of the snow depth stations have neither temperature or precipitation measurements, we construct these time series using monthly homogenized values of temperature and precipitation using nearby stations over the period 1901-2021.

First, we obtain November-March sums of precipitation and averages of mean temperatures at all of homogenized stations (see Figure 1) over the years 1901/02-2020/21. Additionally, time series of standardized anomalies (i.e., z-scores) are computed for each homogenized station (the mean and standard deviations are computed using the calibration period 1902-1971). Then, for each snow depth station, we find the nearest five homogenized stations for temperature and separately for precipitation. Next, we compute two time series. The first contains the averaged standardized anomalies of temperature or precipitation as

a function of the inverse distance of the nearest five stations. The second time series contains the averaged absolute values of temperature or precipitation as a function of the inverse distance of the nearest five stations. The inverse-distance weighted standardized anomalies of the first time series are then adjusted to match the mean and standard deviation of the second time series. This is performed by simply reversing the steps of computing a z-score, where the weighted standardized anomalies, at each station, are first multiplied by the standard deviation of the second time series at the same station, and then we add the

mean of the second time series (also at the same station). This provides us with time series of seasonal (i.e., November-March) mean temperature and precipitation that are located at each one of our 291 snow depth stations.

### 3.3   Sensitivity of Snow Depth to Temperature and Precipitation

Information concerning the bivariate correlations between either mean temperature and snow depth and/or precipitation and snow depth can be useful. However, correlations by themselves do not provide information about the steepness of the slope

between the two variables. For example, given a $1.0°C$ increase in mean temperature or a a 20% increase in precipitation, what would be the expected impact on snow depth? Additionally, how might these expected changes be affected by elevation? Furthermore, what would be the multivariate impact on snow depth given some combination of mean temperature and precipitation changes? And finally, can we apply methods that do not have an underlying assumption of linearity?

In Figures 4a and 4b, one can observe the spatial distribution of the bivariate correlations between snow depth and temper-

ature, and similarly the correlations between snow depth and precipitation. In Figures 4c and 4d, these correlations are shown to vary as a function of elevation. Generally, we find the largest correlations (either positive or negative) at lower elevations for temperature and at higher elevations for precipitation. For example, a station below 500 meters is more likely to see increases/decreases in temperature translate more strongly to decreases/increases in snow depth than for stations at higher elevations. The opposite influence is observed for snow depth and precipitation. Therefore, stations (or regions) at lower elevations are

primarily driven by changes in temperature, while stations at higher elevations are primarily driven by changes in precipitation. These relationships support prior findings such as Morán-Tejeda et al. (2013) and Schöner et al. (2019).

The utility of the information from Figure 4 can be improved in the following ways: 1) Instead of only considering the bivariate statistical relationship between either temperature with snow depth or precipitation with snow depth, we can consider all three variables in a non-linear, multivariate framework. 2) By computing anomalies of the data, we can leverage information

across multiple stations to provide a more robust empirical-statistical relationship.

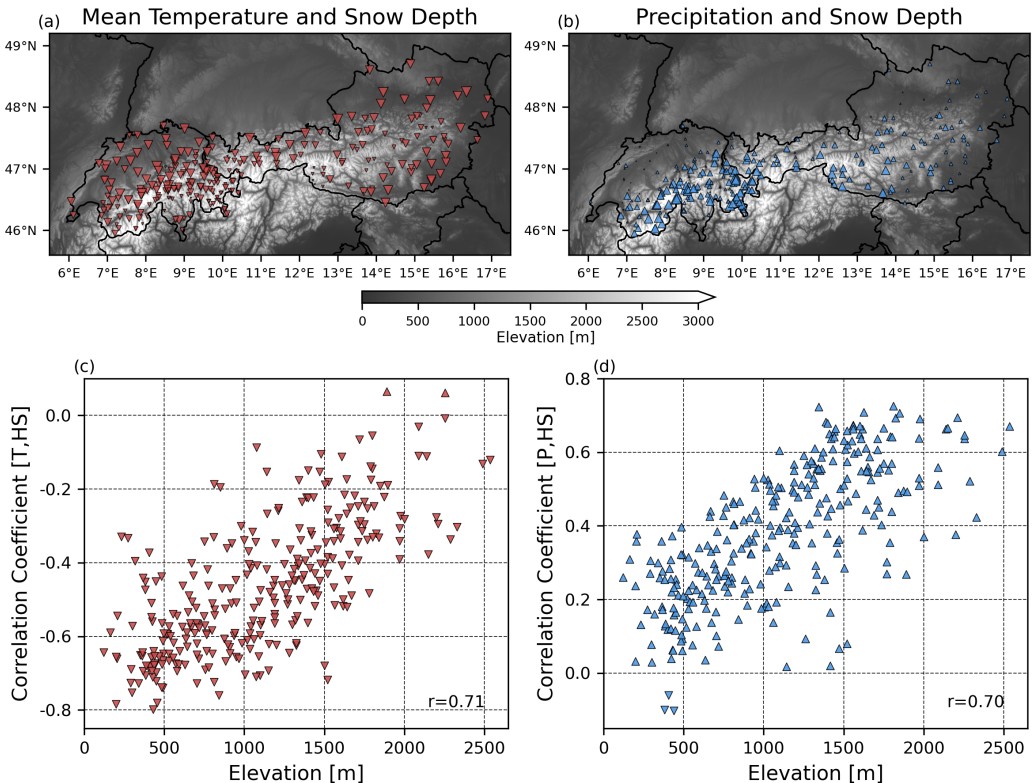

**Figure 4.** The size and magnitudes of the Pearson correlation coefficients between NDJFM snow depth and temperature are plotted in subplot (a). (b) shows the same as (a), but using the seasonal precipitation time series instead of temperature. Downward and upward facing triangles reflect negative and positive correlations, respectively. The sizes of the triangles reflect the magnitude of the correlation. (c) plots the relationship between station elevations and the correlations between snow depth and temperature (i.e., the same values as in (a)), while (d) plots the relationship between station elevations and the correlations between snow depth and precipitation (same values as in (b)).

In Figure 5, we illustrate why it can be important to use anomalous values with our methodology. Figures 5a-5c shows bar plots for the distribution of values using absolute temperature, precipitation, and snow depth, respectively, for the Austrian and Swiss stations between 500-1000 meters. The average station elevation of the Austrian stations used is 745m, while it is 742m for the Swiss stations. Hence, the average station elevations are not much different between the two regions. One can observe in Figs. 5a and 5b that the Swiss stations are generally warmer and wetter than their Austrian counterparts. At the same time, the Swiss stations have lower seasonal averages of snow depth. In order to stress our point regarding the usefulness of anomalies versus absolute values in our methodological context, we can take a further subset of these Austrian and Swiss data points over this 500-1000m elevation band. The individual values of these subsets of data are shown as the scatter plots in Figures 5d-5e. A Student's t-test shows that the means (for temperature, precipitation, and snow depth) of the subset of Austrian data points (Figure 5d) are all statistically significantly different than the subset of Swiss data points (Figure 5e). We find that while

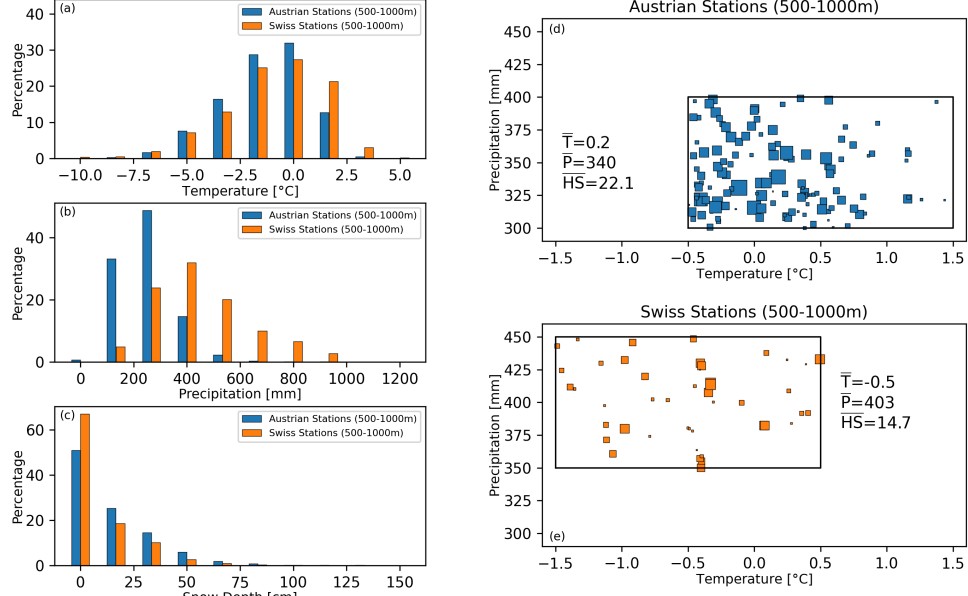

**Figure 5.** Figures 5a-5c show bar plots for the distribution of absolute temperature, precipitation, and snow depth, respectively, for the Austrian and Swiss stations between 500-1000 meters over the historical period 1902-1971. The percentages of the blue and orange bars in each subplot (5a-5c) sum to 100%. The bar plots are comprised of 1,755 observed data points for Austria and 558 data points for Swizterland. A subset of these Austrian and Swiss data points are shown as the scatter plots in 5d and 5e, respectively. These are historical, observed individual seasonal measurements that fall within the climatological regions bounded by the black rectangles. The size of the colored squares reflect the values of absolute snow depth, where larger squares correspond to larger snow depths. The average values of the Austrian and Swiss subsets of the data are provided as the inset text in 5d and 5e.

this subset of data points in Austria has a greater absolute temperature and less absolute precipitation than the Swiss subset, the Austrian stations have significantly more absolute snow depth than the Swiss stations. As we decrease temperature and increase precipitation, we should expect snow depth to increase. However, this is exactly the opposite of what the absolute data is telling us. By simply using the absolute data alone, we can potentially get the wrong signal when comparing one region

to another. This spatial climatological difference can be addressed through computing anomalies, station by station, for the different variables. After computing anomalies, we can then leverage information across a larger region.

To compute anomalies of the data, we begin with absolute (or raw) values of November-March seasonal temperature, precipitation, and snow depth data. Then, November-March average temperature anomalies, $\mathbf{T}^*_{x,t}$, for station $x \in (1, ..., 291)$, and year $t \in (1902, ..., 2021)$, are computed as,

$$\mathbf{T}^*_{x,t} = \mathbf{T}_{x,t} - \overline{\mathbf{T}}_x \ , \tag{4}$$

where $\mathbf{T}_{x,t}$ are the seasonal averages of absolute temperature, and $\overline{\mathbf{T}_\mathbf{x}}$ is the time-averaged mean temperature over the calibration period 1902-1971 at station $x$. Anomalies of November-March precipitation accumulations, $\mathbf{P}^*$, are computed as,

$$\mathbf{P}^*_{x,t} = \frac{\mathbf{P}_{x,t}}{\overline{\mathbf{P}}_x} \quad , \tag{5}$$

where $\mathbf{P}_{x,t}$ are the seasonal accumulations of absolute precipitation, and $\overline{\mathbf{P}}_{\mathbf{x}}$ is the time-averaged precipitation over the period 1902-1971 at station $x$. And similarly, anomalies of November-March average snow depths, $\mathbf{HS}^*$, are obtained by,

$$\mathbf{HS}^*_{x,t} = \frac{\mathbf{HS}_{x,t}}{\overline{\mathbf{HS}}_x} \quad , \tag{6}$$

where $\mathbf{HS}_{x,t}$ are the seasonal averages of absolute snow depth, and $\overline{\mathbf{HS}}_{\mathbf{x}}$ is the time-averaged snow depth over the period 1902-1971 at station $x$.

Once we have computed our data anomlies, we can plot in Figure 6 the observed historical anomalous snow depths (i.e., $\mathbf{HS}^*$) along with temperature and precipitation anomalies (i.e., $\mathbf{T}^*$ and $\mathbf{P}^*$, respectively). The larger squares, bounded by the black lines, correspond to the anomalous measurements for one example station. That station, named "Feldkirch" with number "11110" has the coordinates (lat = 47.27, lon = 9.60) and is situated at an elevation of 439 meters. As one would expect, the average snow depth anomalies increase as the temperature anomaly decreases and the precipitation anomaly increases. This figure can also be used to show that for some increase/decrease in temperature, there can a corresponding increase/decrease in precipitation that will yield approximately the same snow depth anomaly. Consider, for example, the snow depth anomalies in Figure 6 between 2-3 degrees below normal temperature and between 100-150% of normal precipitation. Similar snow depth anomalies can also be observed between 0-1 degrees below normal temperature and between 200-250% of normal precipitation. This gives us a general idea of how sensitive snow depth anomalies are to temperature and precipitation anomalies. Even though Figure 6 gives us a first look at the multivariate sensitivity of snow depth to temperature and precipitation anomalies, we can refine the approach by adding in a third variable. The sensitivities will change as elevation changes. Therefore, we can break up the multivarite sensitivities shown in Fig. 6 into different elevation bands. We have chosen to use four elevation bands: 0-500 meters (containing 52 stations), 500-1000 meters (75 stations), 1000-1500 meters (91 stations), and >1500 meters (73 stations).

The points along the left column in Figure 7 are like those from Figure 6, except that the data is broken up by the four elevation bands, and the data is now only plotted for the calibration period of 1902-1971. This is the data that we will use to fit a model, and make model estimates for our 1972-2021 validation period.

To begin, we calculate averages of snow depth anomalies across 2-dimensional bins of mean temperature and precipitation for different elevation bands. In our effort to construct the sensitivity diagrams, let us first consider the elevation band 0-500 meters. In the lowest elevation band over the calibration period, the arrays $\mathbf{T}^*$, $\mathbf{P}^*$, and $\mathbf{HS}^*$ all have a maximum possible number of measurements of 3,640 (which is 70 seasons, 1902-1971, multiplied by the 52 stations from that band). Next, we find all of the values in $\mathbf{HS}^*_{0-500m}$ that fall within a $0.8°C$ window centered about a given temperature anomaly (with $0.2°C$ increments of the binning window), $\mathbf{T}^*_{0-500m}$, and a 40% window centered about a given precipitation anomaly (with 10% increments), $\mathbf{P}^*_{0-500m}$. We did experiment using different bin sizes, though we found that the choice of bin size does not strongly affect model performance (please refer to our comments in the paper discussion at https://doi.org/10.5194/egusphere-

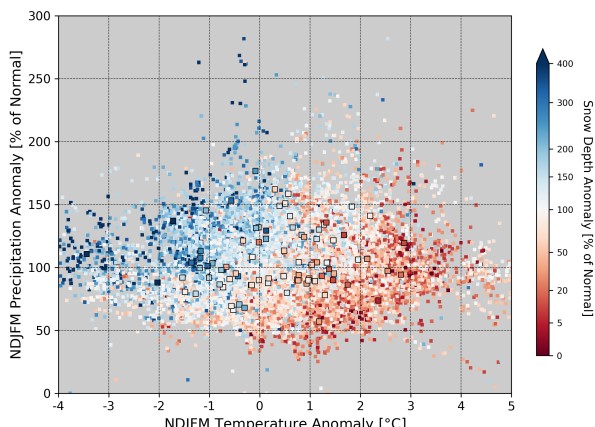

**Figure 6.** Seasonal temperature anomalies, for all stations and all seasons, are plotted on the x-axis against seasonal precipitation anomalies on the y-axis. The colormap corresponds to snow depth anomalies, given the pairings of temperature and precipitation anomalies. That open black squares show the values for one example station, which is named "Feldkirch" with number "11110" and has the coordinates latitude = 47.27 and longitude = 9.60. These anomalies are shown for the entire 1902-2021 period of record.

2024-1172-AC2). We calculate the average of all of the station snow depth anomalies that fall within this 2-dimensional window of temperature and precipitation anomalies, given that there were at least 50 observed snow depth measurements that fall within that 2-dimensional window. Then, we move the center of the window in order to perform the same set of operations across the range of temperature and precipitation anomalies. And lastly, we repeat the process for the other three elevation
bands. The resulting averages using these two-dimensional bins are shown in Figs. 7e-7h.

Next, the snow depth anomalies, resulting from the multivariate binning (Figs. 7e-7h), are used to construct a smoothed and extrapolated sensitivity surface that will then allow us to estimate new values outside of what has been seen in the calibration record. For all binned values across the two-dimensional temperature/precipitation space, distances (in data space) are computed between each specific bin-center and the center of all of the bins where we computed averages in the previous section
(i.e., those are the grid cells which are colored in Figs. 7e-7h). Multiple linear regression is then fit to the nearest quartile of values, where the bin-centers of mean temperature and precipitation anomalies are the predictors and the binned averages of snow depth anomalies are our predictands. To find the nearest quartile of values, we use a Euclidean distance measure which essentially equates the distances of a 10% precipitation anomaly with a 0.2°C temperature anomaly. So, a data point that had the coordinates of (0.4°C warmer, 0% of normal precip) with respect to a point of interest, and another data point with coordi-
nates (0.0°C, 20% of normal precip), would be treated as the same distance. We did not find the model to be overly sensitive to providing more or less weight to the temperature or precipitation axes. Then, the regression coefficients are used along with the the center point of the bin to obtain a snow depth anomaly. Our application of localized linear regression is only fit to the nearest quartile of data points for each bin-center, and therefore, it can accomadate a non-linear response surface across most

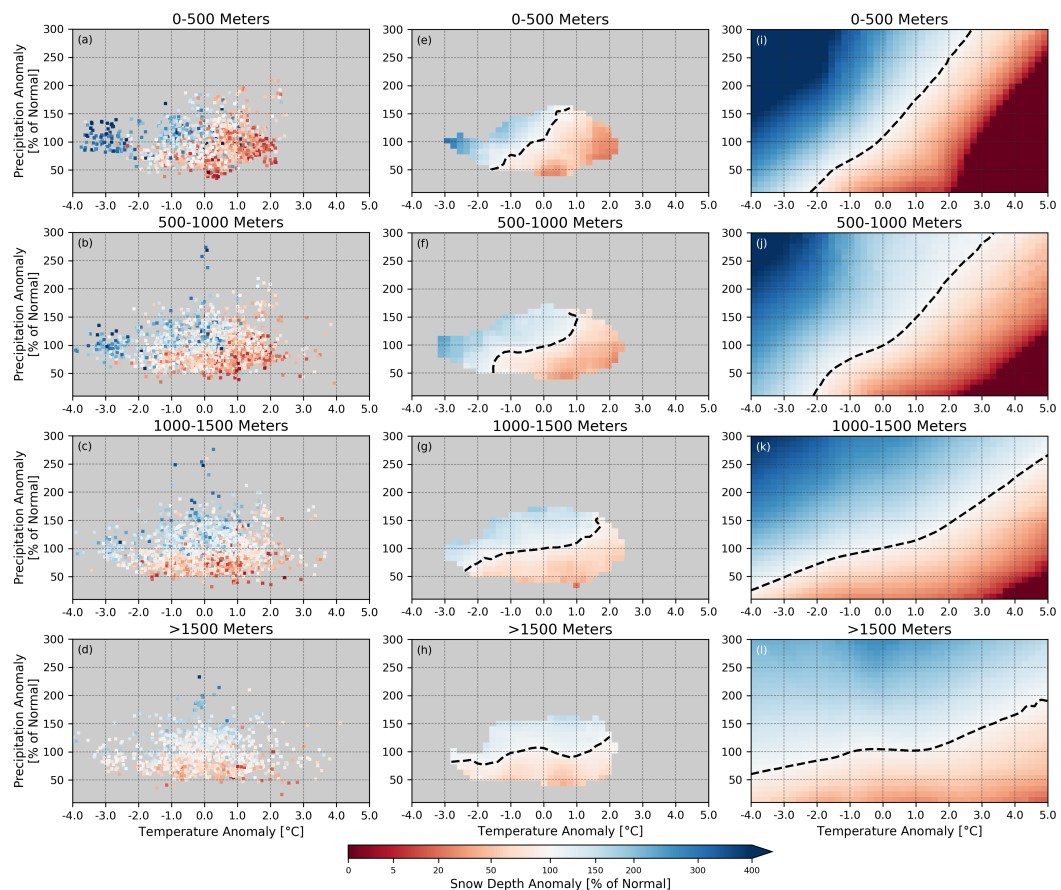

**Figure 7.** Figures 7a-7d and the same as Figure 6, except that the data is broken up by elevation band and we only plot the data over the 1902-1971 calibration period. 7e-7h show the binned anomalies of snow depth using two-dimensional bining windows. The bins are centered using $0.8°C$ temperature binnning window (with $0.2°C$ increments of the binning window) and a 40% precipitation window (with 10% increments). 7i-7l show the fitted surface through localized linear regression using the nearest quartile of bins from 7e-7h (see text). The dashed black lines show contours of 100% of normal snow depth.

of the data domain. At the same time, it smooths out the sensitivity surface (Figs. 7i-7l), and also provides extrapolated values

beyond what was observed in the calibration period.

### 3.4  A Multivariate Sensitivity Model to Estimate Snow Depth

The multivariate sensitivity plots shown in Figs. 7i-7l, which are constructed using only data in the 1902-1971 calibration period, are now used to estimate snow depth anomalies at all stations and for all NDJFM seasons over the period 1972-2021. For each station, first we determine the elevation band in which it falls (e.g., 0-500m, 500-1000m, etc.). Second, we locate the

nearest bin-center given each season's (e.g., November 1971 - March 1972) anomalies of mean temperature and precipitation.

The snow depth anomaly corresponding to that bin is then used as the estimated value for that season for that station. For example, consider a station in the elevation band 0-500m which experienced a mean seasonal temperature 2.0°C above normal (where normal is defined with respect to 1902-1971) and a mean seasonal precipitation at 100% of normal. The snow depth anomaly in Fig. 7i at 2.0°C along the x-axis and 100% of normal along the y-axis corresponds to 37.6% (indicated by the color at that bin). So, given those meteorological anomalies at that station and in that season, the historical sensitivities would estimate that the snow depth would be 37.6% of normal. This procedure is repeated to produce the model estimates for all 291 stations and for all 50 seasons over the 1972-2021 validation period. The model estimates derived in this manner, and driven by the historical sensitivities, are referred to as the SnowSens model for the duration of the paper.

To be clear, we use known values of concurrent seasonal temperature and precipitation anomalies to estimates snow depth using the SnowSens model. As a result, our modeled values are not really forecasts. While we refer to the values produced by the SnowSens model as "estimates," our broader aim is that the model is capable of producing actual forecasts of future snow depth conditions given future projections of temperature and precipitation. True long-term forecasts of snow depth over some future period can be thought of as a modeling chain with two chain links. The first link in the modeling chain are the projections of temperature and precipitation, while the second link are the projections of snow depth driven by the temperature and precipitation projections from the first link. In this paper, our model validation is quantifying the skillfulness or the uncertainty associated with the second link of that modeling chain. With knowledge of how skillful the SnowSens model is, the model can then be forced or driven with a plausible range of future projections of temperature and precipitation. In that way, the SnowSens model can be run to produce actual forecasts of snow depth, which also contain the uncertainty of future projections from the first link of the modeling chain.

When evaluating the performance of the SnowSens and the SNOWGRID-CL models, we use both the absolute and anomalous values. The methodology outlined above provides SnowSens model estimates, as anomalies, for all stations and seasons. At the same time, the SNOWGRID-CL model provides estimates of absolute snow depths. Therefore, we must produce absolute estimated values for the SnowSens model and estimated anomalies for the SNOWGRID-CL model. Let us begin with the estimates of the SnowSens model, were the absolute snow depths, $\mathbf{HS}_{MOD}$, are computed as,

$$\mathbf{HS}_{MOD,x,t} = \mathbf{HS}^*_{MOD,x,t} \times \overline{\mathbf{HS}}_{OBS,x} \ , \tag{7}$$

where $\mathbf{HS}^*_{MOD,x,t}$ are the SnowSens estimated anomalies at station $x$ and time $t$, and $\overline{\mathbf{HS}}_{OBS,x}$ is the observed mean seasonal snow depth at station $x$ over the calibration period 1902-1971. Now, we have absolute estimates for both models. Next, we want to address any mean biases present in the models, while simultaneously computing anomalies of the data. To do this, a common period of record is used. The SNOWGRID-CL model data begins in 1962, and therefore we then use the common reference period 1962-1971 to compute the anomalies and bias correction.

$$\mathbf{HS}^*_{MOD_{BC},x,t} = \frac{\mathbf{HS}_{MOD,x,t}}{\overline{\mathbf{HS}}_{MOD_{1962-1971},x}} \times \overline{\mathbf{HS}}^*_{OBS_{1962-1971},x} \ , \tag{8}$$

where $\mathbf{HS}^*_{MOD_{BC},x,t}$ are our estimated anomalies at station $x$ and time $t$ which have been bias corrected to remove any mean biases present over the common period 1962-1971. Eq. 8 is applied to both models to provide the estimated anomalies which

have been mean bias corrected for all of the stations (i.e., all Austrian stations for SNOWGRID-CL and all Austrian and Swiss stations for SnowSens).

## 4  Results

### 4.1  Model Performance

#### 4.1.1  Comparing the SnowSens and SNOWGRID-CL Models

The skill of the SnowSens model is evaluated with respect to the SNOWGRID-CL model. The SNOWGRID-CL model is run over the Austrian domain, and as a result, the performance of the SnowSens model is evaluated in this section only using the 107 stations within Austria.

Prior to any bias correction, the SNOWGRID-CL estimates perform about as well or worse than climatology. The performance of the SNOWGRID-CL absolute model values over the period 1972-2021 can be evaluated using different periods to calculate the climatological mean, or what is treated as normal. When this period is 1902-1971, the SNOWGRID-CL has an $\mathbf{SS}_{RMSE}$ value of 0.03, and when the 1972-2021 period is treated as the climatological normal, SNOWGRID-CL has an $\mathbf{SS}_{RMSE}$ value of -0.09. In contrast, the absolute estimates of the SnowSens model has greater skill with $\mathbf{SS}_{RMSE}$ values of 0.25 and 0.16, respectively for the two different climatological periods (the skills with respect to climatologies computed over the 1902-1971 period can be found in Table 2).

After applying mean bias correction to both models, the performance of the SNOWGRID-CL model is much improved. Now, using the estimated anomalies which have been bias-corrected, the skills (i.e., $\mathbf{SS}_{RMSE}$) of SNOWGRID-CL and SnowSens models over the 1972-2021 evaluation period are 0.34 and 0.26, respectively. Figures 8a-d show the modeled seasonal snow depth anomalies against the observed seasonal snow depth anomalies for the two models for each of the four elevation bands. While both contain statistically significant skill (p<0.01, where we used bootstrapping to assess the statistical significance), the bias-corrected SNOWGRID-CL model is found to be more skillful than the SnowSens model when it comes to modeling the year-to-year variability of the seasonally averaged snow depths.

Next, we want to know how well the observed trends over the evaluation period have been modeled by both SNOWGRID-CL and SnowSens. The relative changes are computed for all of the Austrian stations between the period 1997-2021 and the period 1972-1996. This is done for the two models, and for the observations. These values are plotted in Figure 8e. When it comes to correctly modeling the trend, the SnowSens model now outperforms SNOWGRID-CL. SNOWGRID-CL generally overestimates the changes over the last 50 years. The skill scores, $\mathbf{SS}_{RMSE}$, for the modeled versus observed changes over the evaluation period are 0.19 for the SnowSens model and 0.10 for SNOWGRID-CL (see Table 2 for a number of comparative skill scores).

We also computed an ensemble average of the two models using their estimated anomalies, and this was found to be more skillful than either model alone. This is true for both the skill scores of year-to-year variability and relative changes observed

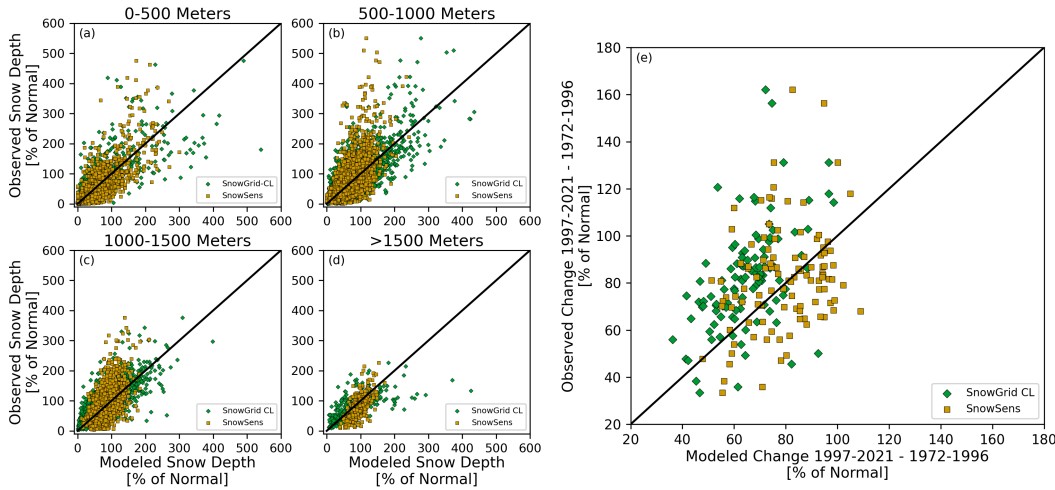

**Figure 8.** SnowSens and SNOWGRID-CL estimated seasonal (Nov-Mar) snow depth anomalies (x-axis) plotted against observations (y-axis) for stations which fall in each of the four elevation bands, (a)-(d), respectively. The estimated changes in seasonal snow depth for the two models are plotted against observed changes in (e). Changes in (e) are the differences, at each station, in the anomalous seasonal snow depths between the more recent period 1997-2021 and the prior period 1972-1996. The black lines in (a)-(e) show the one-to-one lines.

|  | $0-500m$ | $500-1000m$ | $1000-1500m$ | $>1500m$ | All Stations |
|---|---|---|---|---|---|
| SNOWGRID-CL $\mathbf{SS}_{RMSE}$ (**HS**) | -0.24 | 0.05 | -0.06 | 0.18 | 0.03 |
| SnowSens $\mathbf{SS}_{RMSE}$ (**HS**) | 0.44 | 0.21 | 0.26 | 0.25 | 0.25 |
| SNOWGRID-CL $\mathbf{SS}_{RMSE}$ (**HS***) | 0.37 | 0.35 | 0.34 | -0.02 | 0.34 |
| SnowSens $\mathbf{SS}_{RMSE}$ (**HS***) | 0.39 | 0.20 | 0.22 | 0.27 | 0.26 |
| Ensemble Mean $\mathbf{SS}_{RMSE}$ (**HS***) | 0.47 | 0.35 | 0.37 | 0.31 | 0.39 |
| SNOWGRID-CL $\mathbf{SS}_{RMSE}$ ($\%Changes$) | 0.37 | -0.10 | 0.05 | 0.41 | 0.10 |
| SnowSens $\mathbf{SS}_{RMSE}$ ($\%Changes$) | 0.35 | 0.05 | 0.24 | 0.15 | 0.19 |
| Ensemble Mean $\mathbf{SS}_{RMSE}$ ($\%Changes$) | 0.39 | 0.04 | 0.35 | 0.62 | 0.24 |

**Table 2.** A comparison of different skill scores for the two models over the Austrian domain. In the top two rows, the absolute year-to-year skill (i.e., using the raw model estimates) is shown for each elevation band and for all stations. The middle three rows give the year-to-year model skill using the estimated anomalies, which have been bias-corrected, for each of the two models and an ensemble average of the two. The last three rows provide skill scores of how well the models estimate relative changes between the period 1997-2021 and the period 1972-1996.

in the last 50 years. Given this result, using an ensemble such as between these two models has the potential to futher improve the projections of future seasonal snow depth.

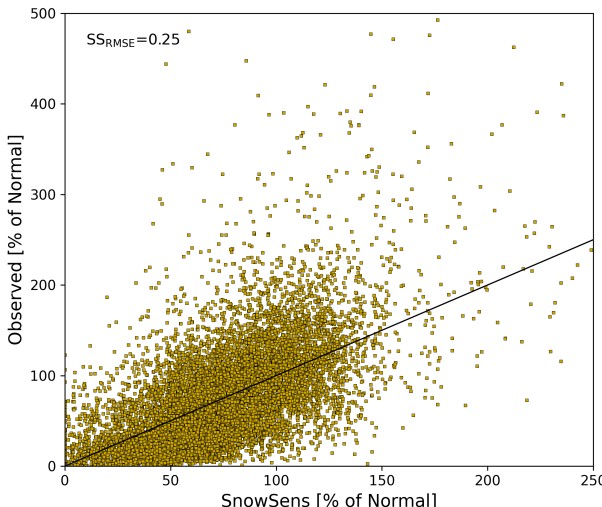

**Figure 9.** All seasonal SnowSens model estimates versus observations over the validation period using all of the stations in the study domain. Again, the black line plots the one-to-one line.

### 4.1.2 Model Performance for the Entire Domain

In the last section, we compared how well the estimates of the SnowSens and SNOWGRID-CL models performed at the locations of the 107 stations across Austria. We investigated the ability of the models to capture both the year-to-year variability

and the historical trends of the observed records. Here, we present the results of the SnowSens model performance for all of the stations. Since we do not have SNOWGRID-CL estimates for the Swiss domain, we are now only showing the results for the SnowSens model.

In the previous section, we found that the SNOWGRID-CL model better captures the observed year-to-year variability at the station scale. However, we want to be clear that the SnowSens model still exhibits substantial skill for the year-to-year seasonal

estimates at the station scale. Figure 9 shows all of the estimated seasonal snow depth anomalies against the observed anomalies for the 1972-2021 validation period. The Pearson correlation coefficient between estimated and observed anmomalies for these 10,985 cases is 0.59, with a $\mathbf{SS}_{RMSE}$ of 0.25 (p<0.01). We used bootstrapping to assess the level of statistical significance, where different shuffled seasons from the calibration period are used as estimates for the validation period. That way, the spatial autocorrelative structure of the seasonal snow depths is preserved in our bootstrapping procedure. We evaluated the skill of

10,000 bootstrapped simulations, and found the largest $\mathbf{SS}_{RMSE}$ value from these randomly generated simulations to be 0.022 (compared to the 0.25 value from the SnowSens model). While we present our p-value as less than 0.01, it is actually much smaller. Please also refer to Table 3 for some additional model skill scores using all of the seasons and all of the stations in study domain.

|  | $0-500m$ | $500-1000m$ | $1000-1500m$ | $>1500m$ | All Stations |
|---|---|---|---|---|---|
| SnowSens $\mathbf{SS}_{RMSE}$ (**HS**) | 0.43 | 0.24 | 0.22 | 0.19 | 0.21 |
| SnowSens $\mathbf{SS}_{RMSE}$ (**HS**$^*$) | 0.32 | 0.24 | 0.20 | 0.20 | 0.25 |
| SnowSens $\mathbf{SS}_{RMSE}$ ($\%Changes$) | 0.49 | 0.10 | 0.21 | -0.02 | 0.24 |

**Table 3.** Skill scores of the SnowSens model for different elevation bands over the entire domain. The top and middle rows are the skill scores using the absolute and estimated anomalies, respectively. The bottom row is the skill in estimating the relative changes between the 1997-2021 and 1972-1996 periods. The three skill scores in the right-most column (which includes all stations over the entire domain) are statistically significant with p<0.01.

We would also like to know how well the model performs in its ability to extrapolate to new climatological terrain. Put another way, how well does the model perform in cases which it hadn't really seen before in the calibration period? In Figure 10, we test the effectiveness of the SnowSens model in a climatological region that had rarely seen observations in the calibration period. Figure 10a shows the 95 cases where the average seasonal temperature in the validation period was greater than +1.0°C and less than 50% of normal precipitation. The error of the SnowSens estimates, for these cases, is less than half of the climatological estimates (indicated by $\mathbf{SS}_{RMSE}$>0.50). The $\mathbf{SS}_{RMSE}$ for the points in Fig. 10a is 0.62, which is statistically significant with p<0.01. Note that nearly all of the modeled and observed values fall below 100% of normal. Averaging across this set of cases (the larger green squares), we find that both the modeled average and the observed average are 33% of normal. In Figure 10b, we increase the sample size by using a threshold of less than 75% of normal precipitation (instead of 50%). This gives us 988 cases. The $\mathbf{SS}_{RMSE}$ in this case is 0.55, which is also found to be statistically significant with p<0.01. The modeled and observed averages over these cases are 42% and 41%, respectively. So, while we are extrapolating to "unknown" climatological terrain, we find the model is quite capable of performing skillfully in that new terrain, especially when aggregating over a number of cases. We do advise, however, that one proceeds with caution when interpreting any of the individual model estimates.

A spatial plot showing the geographical distribution of the station-by-station skill scores is shown in Figure 11a. With the help of Figure 11b, it can be observed that the skill of the model estimates generally decreases with elevation (it is not shown, but the SNOWGRID-CL model also sees decreasing skill with increasing elevation). This makes sense, given that stations at lower elevations are more sensitive to temperature changes, and the range of these temperature changes at lower elevations have been particularly large in the context of climate change. Bootstrapping is used again to test for statistical significance of the model skill at each individual station (Figs. 11a and 11b). We find that there are only a total of 11 stations which either perform worse than climatology or worse than randomly estimated time series (with p<0.01). Therefore, we find that the SnowSens estimates of snow depths from more than 95% of the stations in our study are found to exhibit statistically significant positive skill. Figure 11c plots the modeled and observed relative changes in seasonal snow depth between the periods 1972-2021 and 1902-1971, while Figure 11d shows this between the periods 1997-2021 and 1972-1996. The yellow diamonds are

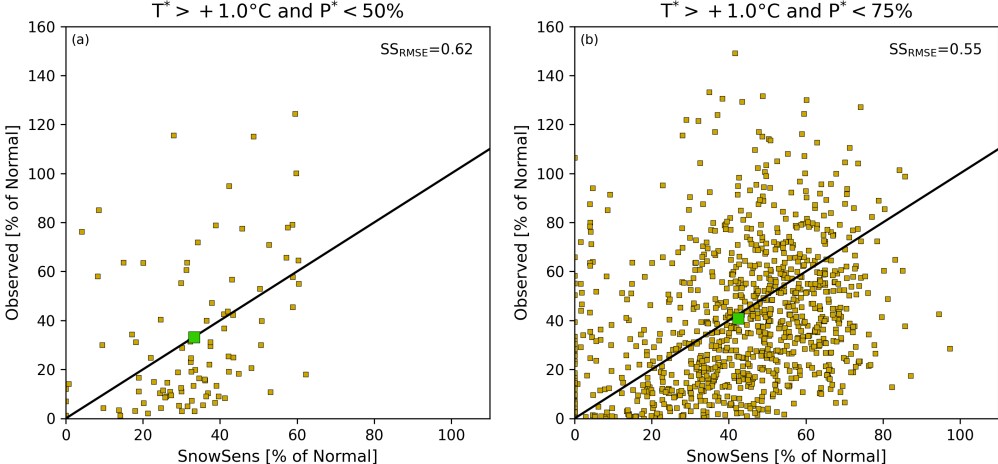

**Figure 10.** Figure 10 shows the effectiveness of the SnowSens model in its ability to estimate in new climatological terrain. (a) plots the pairings of modeled versus observed anomalies for the 95 cases where the average seasonal temperature in the validation period was greater than $+1.0°C$ and less than 50% of normal precipitation. (b) plots the pairings for the 988 cases when using a threshold of less than 75% of normal precipitation (instead of 50%). The larger green squares show the modeled versus observed averages across the respective cases from (a) and (b). Again, the one-to-one lines are plotted in black.

relative changes averaged across the four elevation bands. The $SS_{RMSE}$ of the band-averaged estimates changes for these two periods are 0.80 and 0.73, respectively. By implementing spatial averaging of the estimates across elevation bands, the skill dramatically improves. We should also note that we had also applied the same methodology to snow water equivalent (SWE) values that we constructed via the approach outlined in Winkler et al. (2021). While not shown here, the skill of the SWE estimates follows very closely to the skill in estimating snow depth.

In Figure 12, estimated anomalies of seasonal snow depth are averaged across all of the stations which fall within each of the four elevation bands. The resulting band-averaged time series can be seen alongside the band-averaged observed time series in the subplots of Figure 12. Again, we find that SnowSens is not fully capturing the observed year-to-year variability and is not able to reproduce the high and low extreme values. However, we do find that the model performance is further improved using these band averages.

Figure 13 provides a useful and simple to interpret plot of expected future changes in snow depth as a function of temperature and precipitation anomalies. A user can take a range of expected future projections of temperature and precipitation, and evaluate how these would translate into expected changes in snow depth. These expected changes to snow depth, given different temperature and precipitation scenarios, can be compared to prior studies such as Schmucki et al. (2014) and Marty et al. (2017a). In Figure 13, we have simplified the information provided in Figures 7i-7l. Figure 13 shows cross-sections of Figures 7i-7l for three different precipitation anomalies, and those are 100% of normal along with 80% and 120% of normal (i.e., 20% below and above average). The vertical pink lines show how much warming has already taken place in each elevation band over the period 1972-2021 with respect to the period 1902-1971. The average estimated and observed snow depth anomalies

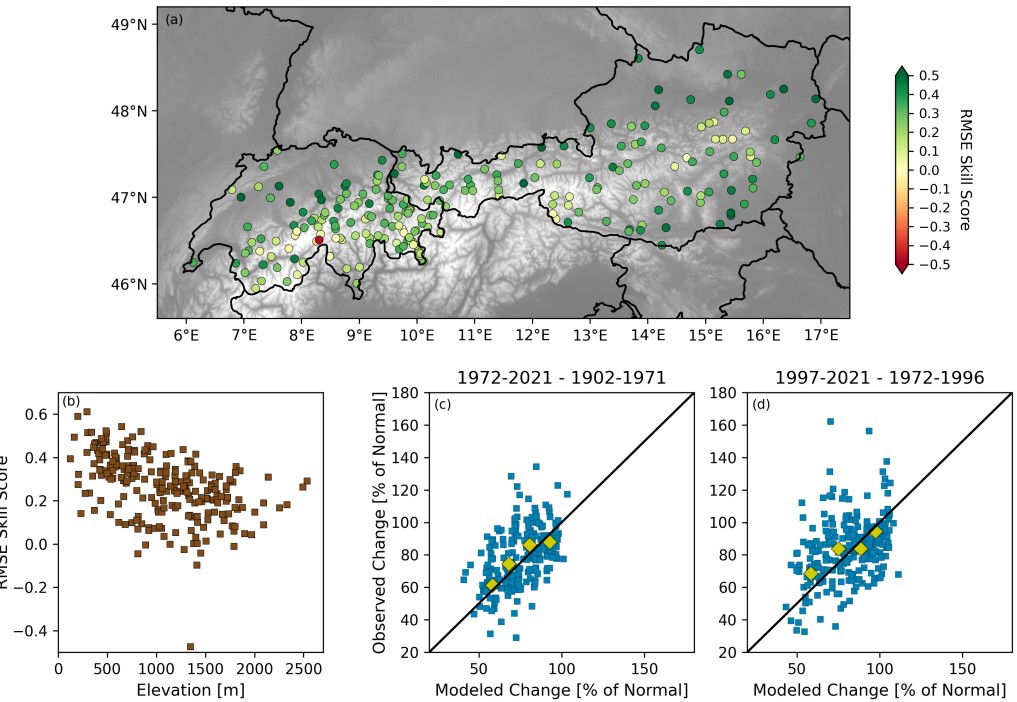

**Figure 11.** (a) Skill over the 1972-2021 evaluation period at each of stations. (b) Plotting how the skill varies with respect to elevation. (c) Modeled versus observed relative changes in snow depth between the periods 1972-2021 and 1902-1971. (d) Same as (c), but using the periods 1996-2020 and 1972-1996. The larger yellow diamonds in (c) and (d) show the changes averaged across the four elevation bands.

are plotted respectively as the open square and the "x". One can use this plot to gain a more detailed understanding of how something like an additional 2°C would translate to snow depth anomalies at different elevations, given the assumption that precipitation stays about the same (100% of the 1902-1971 normal). For elevations below 500 meters, an additional 2°C (which is 3.2°C above the 1902-1971 normal) could lead to seasonally averaged accumulated snow depths being a thing of the past. Put another way, there would be nearly no snow depth accumulation projected at those temperature anomalies. Given two additional degrees of warming in the other three elevation bands, we could expect snow depth anomalies of approximately 25%, 50%, and 80% of normal, respectively.

## 5   Conclusions

Climate change has already had an observable impact on the average seasonal snow depths across the European Alps. Over the historical period 1902-2021, stations across Austria and Switzerland have shown a decrease of seasonally averaged November-March snow depth ranging between -8.1% per decade at elevations below 500 meters and -3.4% per decade for elevations above 1500 meters. Over the more recent historical period 1952-2021, these changes are even greater with decreases ranging

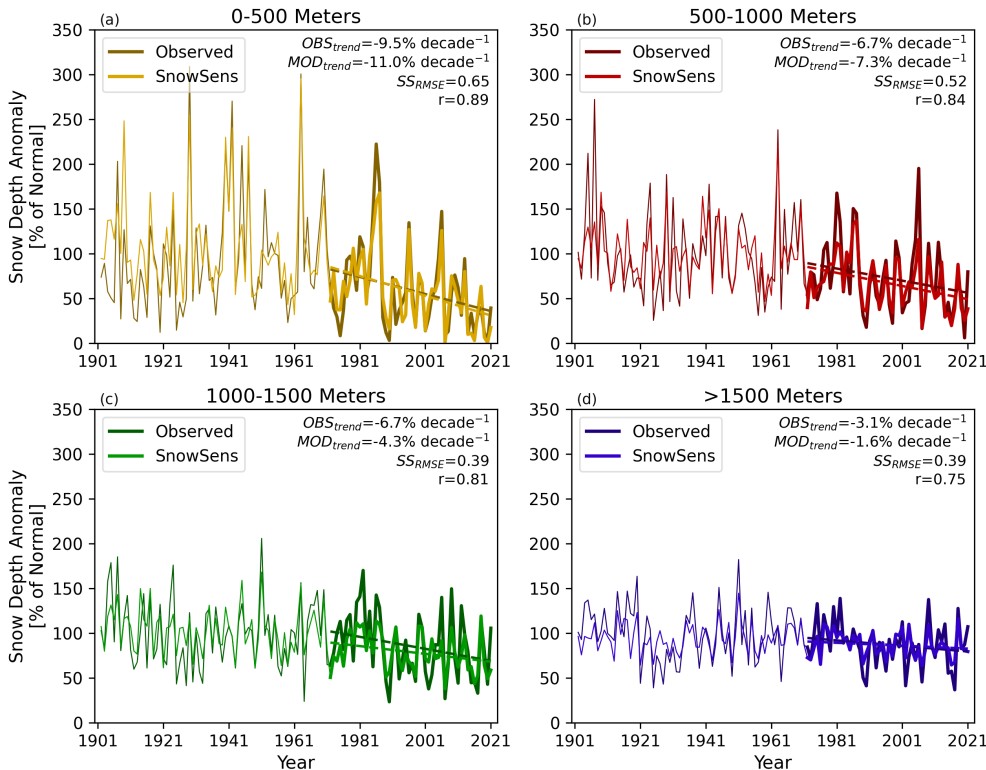

**Figure 12.** Average (Nov-Mar) estimated snow depth anomalies and observations are plotted as time series for each of the four elevation bands. The thinner and thicker lines show the average anomalies during the calibration and validation periods, respectively. The text in the upper right of each subplot lists a select number of metrics (i.e., observed trend per decade, modeled trend per decade, $SS_{RMSE}$, and Pearson correlation coefficient) corresponding to the 1972-2021 validation period. We should also note what the corelation coefficients are between the modeled and observed time series when the trends of both time series have been removed over the validation period. In those cases, the correlations are 0.89, 0.83, 0.79, and 0.74, respectively, which are very similar to what are reported in the figure with the trends present. The number of stations used to compute the band averages are 52, 75, 91, and 73, respectively (see Table 1).

between -15.7% per decade and -3.6% per decade for stations below 500 meters and above 1500 meters, respectively (see Appendix A). Changes in seasonally averaged snow depth can primarily be attributed to changes in meteorological forcing
variables such as mean temperature and precipitation. In some cases, blowing wind and sublimation of snow can greatly affect the snowpack, and in these instances temperature and precipitation alone may not suffice as predictors. In this paper, however, we focus exclusively on using anomalies of temperature and precipitation across different elevation bands as predictors of seasonally averaged snow depth.

Using historical observations of seasonally averaged temperature, precipitation, and snow depth at four different elevation
bands over the period 1902-1971, we constructed a multivariate empirical-statistical model, which is named SnowSens. Model

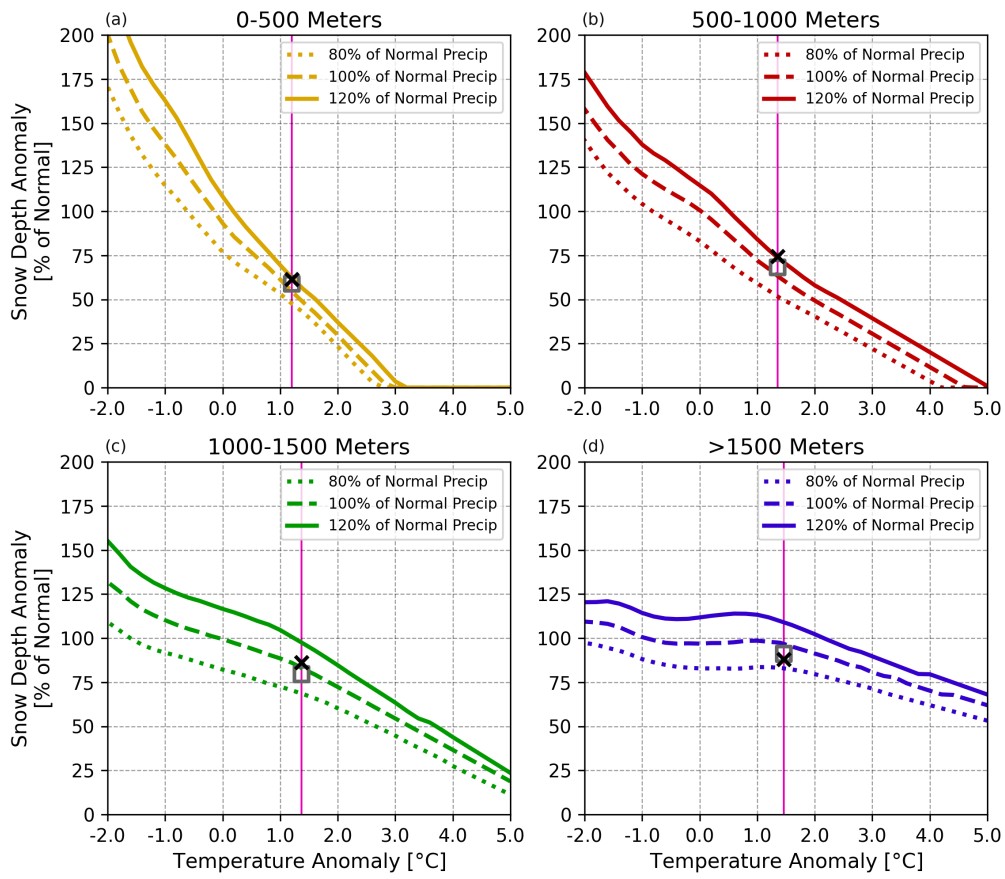

**Figure 13.** This figure plots cross sections of Fig. 7 at 80%, 100% and 120% of normal precipitation. Within a climate change context, this figure serves to provide greater ease in quantifying the changes in snow depth given a range of projected changes in temperature and precipitation. The vertical pink lines show how much warming has already taken place in each elevation band over the period 1972-2021 with respect to the period 1902-1971. The average (Nov-Mar) estimated and observed snow depth anomalies over the 1972-2021 validation period are plotted respectively as the open square and the "x".

validation, which was performed over the period 1972-2021, show that both the SnowSens and SNOWGRID-CL models can skillfully estimate year-to-year seasonally average snow depths across the Austrian domain. While SNOWGRID-CL is found to perform better in estimating year-to-year variability of snow depth, the SnowSens model better estimates historical trends. The SnowSens model is not to be seen as a replacement for operational models such as the SNOWGRID-CL. Rather, this paper

highlights how effectively historical senstivities can be used in a multivariate framework to produce quite accurate estimates of long-term changes in snow depth. The model performs particularly well when the estimates are aggregated over a number of cases, such as across different elevation bands (as seen in Fig. 12). Furthermore, SnowSens relies on a comparatively simplified

modeling framework, which lends itself well to easily translating projected changes in temperature and precipitation to changes in snow depth. Our results show that the historical sensitivities have been robust and persistent. If these sensitivities continue to remain persistent into the future, and future projections of temperature and precipitation are also skillful, then this modeling approach can be expected to yield skillful forecasts for the next 50 years.

The impacts of a changing climate will vary from region to region. We developed multivariate sensitivities that are regionally specific to the Austrian and Swiss Alps. While outside of the scope of this study, the same approach can be applied to other mountain regions. How might the sensitivities of the Rocky Mountains or the Cascades of the United States compare to what we observe in the Alps? Another way our research can be extended relates to quantifying the changes in snow depth versus streamflow. Are specific reductions in snow depth at certain elevations noticeably affecting aggregated streamflow measurements, or is it rather the timing of discharge that is impacted? And lastly, it would be valuable to investigate how capable a variety of different GCMs/RCMs are in capturing the observed sensitivities that we have produced.

Snow depth is a valuable resource that affects many communities adjacent to and downstream of mountain regions. Changes in snow depth can have broad impacts that range from water resources to snow tourism and avalanche preparedness. Climate change is expected to bring about further increases in temperature across the Alps, while it is less clear what the impact will be on precipitation. With improved tools, we can better quantify the impact that these meteorological changes will have on snow depth. Thus, allowing communities to better plan and prepare for the changes to come.

*Data availability.* Supporting data can be found at https://doi.org/10.6084/m9.figshare.25623714.

**Appendix A: Appendix A**

In Figure A1, we plot the historical trends over the more recent period 1952-2021. There are greater relative decreases in snow depth over a more recent historical record 1952-2021 (in contrast to Fig. 3). In this more recent period, we observe decreases in snow depths ranging between -15.7% per decade (Figure A1a) for stations below 500 meters to -3.6% per decade for stations at elevations above 1500 meters (Figure A1d). Anthropogenic climate change is often more clearly recognisable in the recent past, and hence, the trends derived for this period can help to improve our understanding of the expected future changes in snow depth.

Figure A2 plots the raw changes in snow depth over the last 30 year period 1992-2021, compared to the period 1962-1991, as a function of elevation. Greater absolute changes in snow depth are observed as elevation increases. However, this information needs to be placed in the context of differences in climatology. As was shown in Figure 2c, the average seasonal snow depth scales with elevation. The relative changes of snow depth over the last 30 years are plotted against elevation in Figure A2b. Using anomalies, we obtain a stronger relationship between the more recent changes in snow depth and elevation (compare the correlations between Figs. A2a and A2b). The variance explained between elevation and relative changes is approximately 34%, while it is only about 10% when using the absolute changes. In addition to giving us a better statistical relationship,

we also get a clearer picture of where we can observe the greatest relative changes in snow depth. While stations at lower elevations saw smaller absolute changes in their snow depth over the last 30 years, these same stations saw a greater relative decrease over the same period of time. For example, stations below 500 meters had an average absolute change of -1.58 cm, but this reasonably small absolute amount reflected a large relative change, which was 71% of normal. In contrast, stations above 1500 meters had an average absolute change of -8.38 cm, while the average relative change was 93% of normal. Next, we observe whether where changes in snow depth have been greatest/least as a function of geographic location. To do this, we first isolate the influence of latitude and longitude (i.e., x and y space) by removing the dependence of these snow depth changes on elevation (i.e., z space). The dashed line in Figure A2b shows the fitted line through least-squares linear regression of the data. The data are detrended with respect to the regression line, while preserving the population mean (Fig. A2c). The downward-facing and upward-facing triangles in Figure A2d show anomalous snow depth conditions over the period 1992-2021, where the influence of elevation has been removed. Figure A2d, along with Figures A2e and A2f, shows that as one traverses the Austrian and Swiss Alps from east to west and from north to south, the stations experienced slightly greater relative decreases in snow depth.

*Author contributions.* The study was conceived by Matthew Switanek. Gernot Resch and Christoph Marty curated and shared the snow depth data, while Daniel Günter helped to access the SNOWGRID-CL data. All analysis, results, and figures were produced by Matthew Switanek with input from all coauthors. The original draft was written by Matthew Switanek with assistance from all of the other coauthors.

*Competing interests.* The authors do not have any competing interests.

*Acknowledgements.* The authors want to thank the Austrian Water Budget Department for the access to their datasets. This research was developed with the financial support of the FWF (FWF Fonds zur Förderung der wissenschaftlichen Forschung, Project Hom4Snow, Grant Number: I 3692) and the University of Graz. The lead author would like to thank Peter Troch for some early discussions concerning the utility of historical hydrometeorological sensitivities. The lead author would also like to thank Sven Kotlarski (MeteoSwiss) who provided some useful discussion and feedback at an early stage of this study. The authors would also like to thank Michael Matiu and another anonymous reviewer for their help in improving the quality of our paper.

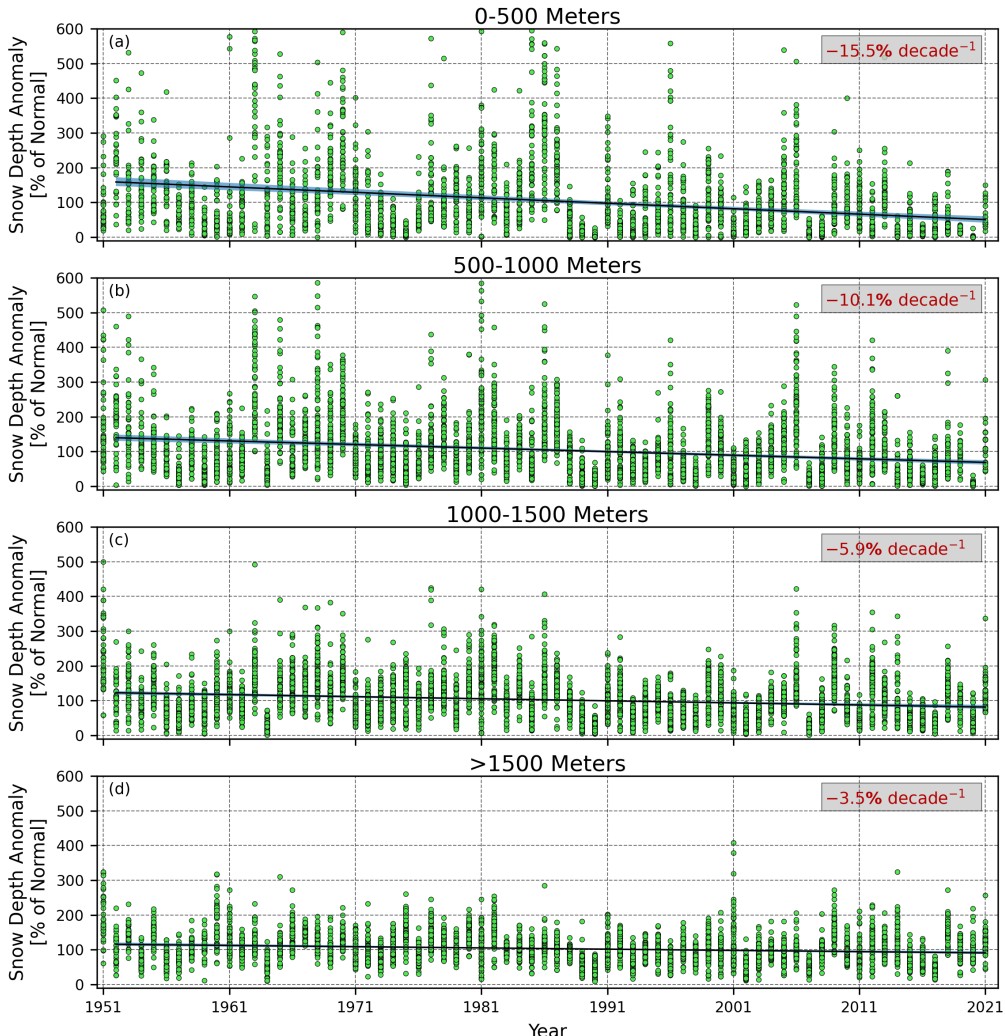

**Figure A1.** Same as Fig. 3, but using the 1952-2021 period of record.

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

480

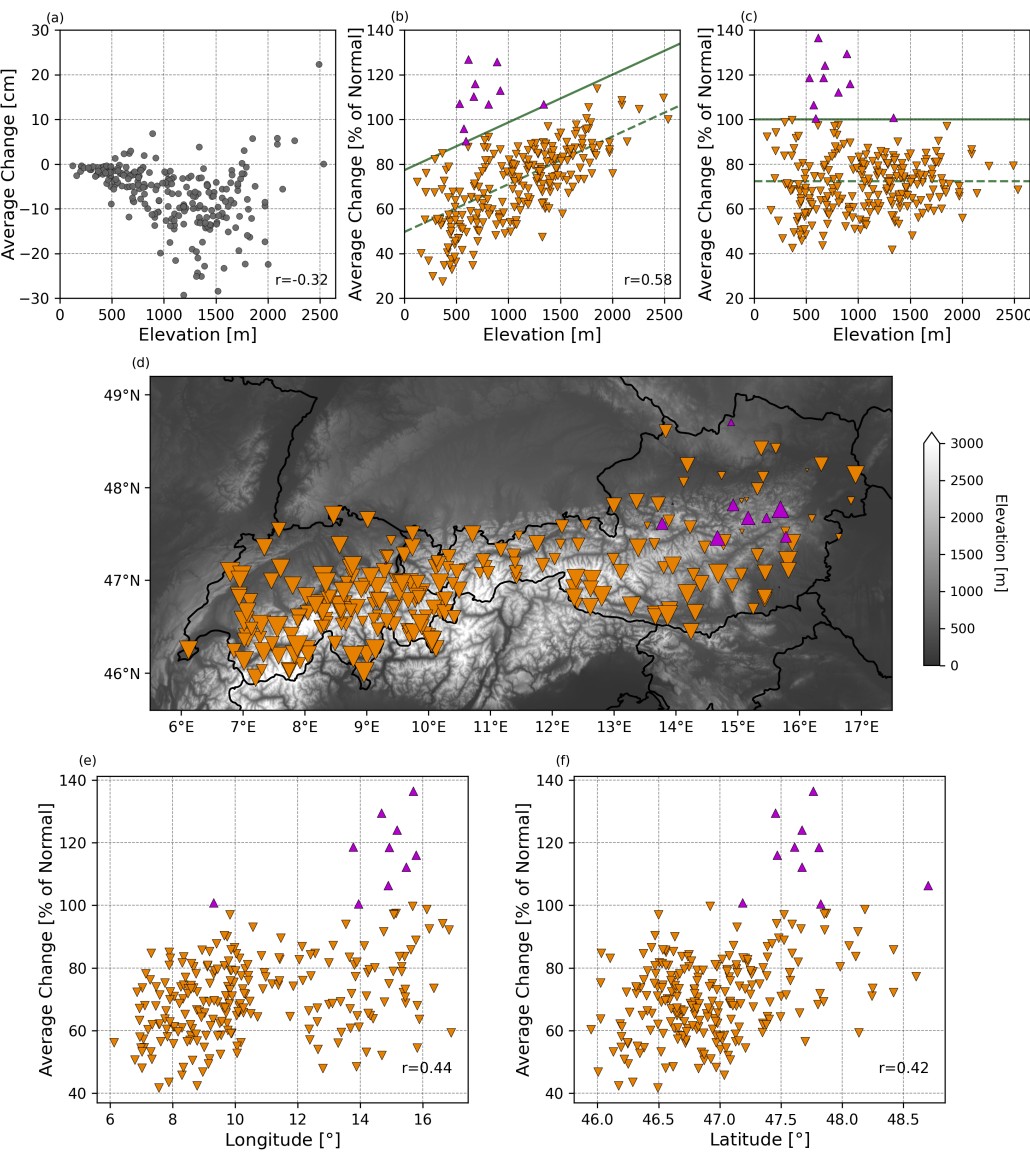

**Figure A2.** Changes in snow depth (1992-2021 vs 1962-1991). (a) shows the absolute snow depth changes as a function of elevation. (b) shows the relative or anomalous snow depth changes as a function of elevation. The dashed line is the fitted least-squares regression of the data. (c) plots the snow depth anomalous changes where the elevation dependence has been removed by detrending the data with respect to the dashed line from (b). The solid lines in (b) and (c) show 100% of normal with the elevation dependence removed, while the colors and direction of the triangles reflect the stations which experienced positive or negative changes after removing elevation dependence. (d) plots the detrended anomalous changes (from Fig. 8c) across the study region. The size of the triangles reflect the size of the anomalies. (e) and (f) show the detrended anomalous changes against longitude and latitude, respectively.

Beniston, M., Farinotti, D., Stoffel, M., Andreassen, L. M., Coppola, E., Eckert, N., Fantini, A., Giacona, F., Hauck, C., Huss, M., Huwald, H., Lehning, M., López-Moreno, J.-I., Magnusson, J., Marty, C., Morán-Tejéda, E., Morin, S., Naaim, M., Provenzale, A., Rabatel, A., Six, D., Stötter, J., Strasser, U., Terzago, S., and Vincent, C.: The European mountain cryosphere: a review of its current state, trends, and future challenges, The Cryosphere, 12, 759–794, https://doi.org/10.5194/tc-12-759-2018, 2018.

Bertoldi, G., Bozzoli, M., Crespi, A., Matiu, M., Giovannini, L., Zardi, D., and Majone, B.: Diverging snowfall trends across months and elevation in the northeastern Italian Alps, Int. J. Climatol., 43, 2794–2819, https://doi.org/10.1002/joc.8002, 2023.

Buchmann, M., Coll, J., Aschauer, J., Begert, M., Brönnimann, S., Chimani, B., Resch, G., Schöner, W., and Marty, C.: Homogeneity assessment of Swiss snow depth series: comparison of break detection capabilities of (semi-)automatic homogenization methods, The Cryosphere, 16, 2147–2161, https://doi.org/10.5194/tc-16-2147-2022, 2022.

Colombo, N., Guyennon, N., Valt, M., Salerno, F., Godone, D., Cianfarra, P., Freppaz, M., Maugeri, M., Manara, V., Acquaotta, F., Petrangeli, A. B., and Romano, E.: Unprecedented snow-drought conditions in the Italian Alps during the early 2020s, Environmental Research Letters, 18, 074 014, https://doi.org/10.1088/1748-9326/acdb88, 2023.

Elsasser, H. and Bürki, R.: Climate change as a threat to tourism in the Alps, The Cryosphere, 20, 253–257, 2002.

Gobiet, A., Kotlarski, S., Beniston, M., Heinrich, G., Rajczak, J., and Stoffel, M.: 21st century climate change in the European Alps - A review, Sci. Total Environ., 493, 1138–1151, https://doi.org/10.1016/j.scitotenv.2013.07.050, 2014.

Haberkorn, A.: European Snow Booklet - an Inventory of Snow Measurements in Europe, https://doi.org/10.16904/envidat.59, 2019.

Hiebl, J. and Frei, C.: Daily temperature grids for Austria since 1961 - concept, creation and applicability, Theoretical and applied climatology, 124, 161–178, https://doi.org/10.1007/s00704-015-1411-4, 2016.

Hiebl, J. and Frei, C.: Daily precipitation grids for Austria since 1961 - development and evaluation of a spatial dataset for hydroclimatic monitoring and modelling, Theoretical and applied climatology, 132, 327–345, https://doi.org/10.1007/s00704-017-2093-x, 2018.

Hüsler, F., Jonas, T., Riffler, M., Musial, J. P., and Wunderle, S.: A satellite-based snow cover climatology (1985–2011) for the European Alps derived from AVHRR data, The Cryosphere, 8, 73–90, https://doi.org/10.5194/tc-8-73-2014, 2014.

Laternser, M. and Schneebeli, M.: Long-term snow climate trends of the Swiss Alps (1931–99), International Journal of Climatology, 23, 733–750, 2003.

Lievens, H., Demuzere, M., Marshall, H. P., Reichle, R. H., Brucker, L., Brangers, I., de Rosnay, P., Dumont, M., Girotto, M., Immerzeel, W. W., and Jonas, T.: Snow depth variability in the Northern Hemisphere mountains observed from space, Nat. Commun., 10(1), 4629, https://doi.org/10.1038/s41467-019-12566-y, 2019.

Lievens, H., Brangers, I., Marshall, H.-P., Jonas, T., Olefs, M., and De Lannoy, D.: Sentinel-1 snow depth retrieval at sub-kilometer resolution over the European Alps, The Cryosphere, 16, 159–177, https://doi.org/10.5194/tc-16-159-2022, 2022.

Marcolini, G., Koch, R., Chimani, B., Schöner, W., Bellin, A., Disse, M., and Chiogna, G.: Evaluation of homogenization methods for seasonal snow depth data in the Austrian Alps, 1930–2010, Int. J. Climatol., 39, 4514–4530, https://doi.org/10.1002/joc.6095, 2019.

Marty, C.: Regime shift of snow days in Switzerland, Geophysical Research Letters, 35, https://doi.org/10.1029/2008GL033998, 2008.

Marty, C. and Blanchet, J.: Long-term changes in annual maximum snow depth and snowfall in Switzerland based on extreme value statistics, Climatic Change, 111, 705–721, https://doi.org/10.1007/s10584-011-0159-9, 2012.

Marty, C., Schlögl, S., Bavay, M., and Lehning, M.: How much can we save? Impact of different emission scenarios on future snow cover in the Alps, The Cryosphere, 11, 517–529, https://doi.org/10.5194/tc-11-517-2017, 2017a.

Marty, C., Tilg, A.-M., and Jonas, T.: Recent Evidence of Large-Scale Receding Snow Water Equivalents in the European Alps, J. Hydrometeorol., 18, 1021–1031, https://doi.org/10.1175/JHM-D-16-0188.1, 2017b.

Marty, C., Rohrer, M. B., Huss, M., and Stähli, M.: Multi-decadal observations in the Alps reveal less and wetter snow, with increasing variability, Front. Earth Sci., 11, https://doi.org/doi.org/10.3389/feart.2023.1165861, 2023.

Matiu, M., Crespi, A., Bertoldi, G., Carmagnola, C.-M., Marty, C., Morin, S., Schöner, W., Berro, D.-C., Chiogna, G., Gregorio, L. D.,

Kotlarski, S., Majone, B., Resch, G., Terzago, S., Valt, M., Beozzo, W., Cianfarra, P., Gouttevin, I., Marcolini, G., Notarnicola, C., Petitta, M., Scherrer, S., Strasser, U., Winkler, M., Zebisch, M., Cicogna, A., Cremonini, R., Debernardi, A., Faletto, M., Gaddo, M., Giovannini, L., Mercalli, L., Soubeyroux, J.-M., Sušnik, A., Trenti, A., Urbani, S., and Weilguni, V.: Observed snow depth trends in the European Alps: 1971 to 2019, The Cryosphere, 15, 1343–1382, https://doi.org/10.5194/tc-15-1343-2021, 2021.

Monteiro, D. and Morin, S.: Multi-decadal analysis of past winter temperature, precipitation and snow cover data in the European Alps from

reanalyses, climate models and observational datasets, The Cryosphere, 17, 3617–3660, https://doi.org/10.5194/tc-17-3617-2023, 2023.

Morán-Tejeda, E., López-Moreno, J.-I., and Beniston, M.: The changing roles of temperature and precipitation on snowpack variability in Switzerland as a function of altitude, Geophys. Res. Lett., 40, 2131–2136, https://doi.org/10.1002/grl.50463, 2013.

Notarnicola, C.: Overall negative trends for snow cover extent and duration in global mountain regions over 1982–2020, Scientific Reports, 12, https://doi.org/10.1038/s41598-022-16743-w, 2022.

Olefs, M., Schöner, W., Suklitsch, M., Wittmann, C., Niedermoser, B., Neururer, A., and Wurzer, A.: SNOWGRID-A new operational snow cover model in Austria, in: International snow science workshop grenoble-chamonix mont-blanc, pp. 38–45, 2013.

Olefs, M., Koch, R., Schöner, W., and Marke, T.: Changes in Snow Depth, Snow Cover Duration, and Potential Snowmaking Conditions in Austria, 1961–2020—A Model Based Approach, Atmosphere, 11, 1–21, https://doi.org/10.3390/atmos11121330, 2020.

Pepin, N. C., Arnone, E., Gobiet, A., Haslinger, K., Palazzi, E., Seibert, P., Serafin, S., Schöner, W., Vuille, M., and Adler, C.: Climate Changes

and Their Elevational Patterns in the Mountains of the World, Reviews of Geophysics, 60, https://doi.org/10.1029/2020RG000730, 2022.

Reid, P. C., Hari, R. E., Beaugrand, G., Livingstone, D. M., Marty, C., Straile, D., Barichivich, J., Goberville, E., Adrian, R., Aono, Y., Brown, R., Foster, J., Groisman, P., Hélaouët, P., Hsu, H.-H., Kirby, R., Knight, J., Kraberg, A., Li, J., Lo, T.-T., Myneni, R. B., North, R. P., Pounds, J. A., Sparks, T., Stübi, R., Tian, Y., Wiltshire, K. H., Xiao, D., and Zhu, Z.: Global impacts of the 1980s regime shift, Global change biology, 22, 682–703, https://doi.org/10.1111/gcb.13106, 2016.

Resch, G., Koch, R., Marty, C., Buchmann, M., Begert, M., and Schöner, W.: A quantile-based approach to improve homogenization of snow depth time series, Int. J. Climatol., 43, 157–173, https://doi.org/10.1002/joc.7742, 2022.

Scalzitti, J., Strong, C., and Kochanski, A.: Climate change impact on the roles of temperature and precipitation in western U.S. snowpack variability, Geophysical Research Letters, 43, 5361–5369, https://doi.org/10.1002/2016GL068798, 2016.

Scherrer, S. C. and Appenzeller, C.: Swiss Alpine snow pack variability: major patterns and links to local and large-scale flow, Clim. Res.,

32, 187–199, 2006.

Scherrer, S. C., Wüthrich, C., Croci-Maspoli, M., Weingartner, R., and Appenzeller, C.: Snow variability in the Swiss Alps 1864–2009, Int. J. Climatol., 33, 3162–3173, https://doi.org/10.1002/joc.3653, 2013.

Schmucki, E., Marty, C., Fierz, C., and Lehning, M.: Simulations of 21st century snow response to climate change in Switzerland from a set of RCMs, International Journal of Climatology, 35, 3262–3273, https://doi.org/10.1002/joc.4205, 2014.

Schöner, W., Koch, R., Matulla, C., Marty, C., and Tilg, A.-M.: Spatiotemporal patterns of snow depth within the Swiss-Austrian Alps for the past half century (1961 to 2012) and linkages to climate change, Int. J. Climatol., 39, 1589–1603, https://doi.org/10.1002/joc.5902, 2019.

Sippel, S., Fischer, E. M., Scherrer, S. C., Meinshausen, N., and Knutti, R.: Late 1980s abrupt cold season temperature change in Europe consistent with circulation variability and long-term warming, Environ. Res. Lett., 15, https://doi.org/10.1088/1748-9326/ab86f2, 2020.

Sospedra-Alfonso, R., Melton, J. R., and Merryfield, W. J.: Effects of temperature and precipitation on snowpack variability in the Central Rocky Mountains as a function of elevation, Geophysical Research Letters, 42, 4429–4438, https://doi.org/10.1002/2015GL063898, 2015.

Viviroli, D., Archer, D. R., Buytaert, W., Fowler, H. J., Greenwood, G. B., Hamlet, A. F., Huang, Y., Koboltschnig, G., Litaor, M. I., López-Moreno, J. I., Lorentz, S., Schädler, B., Schreier, H., Schwaiger, K., Vuille, M., and Woods, R.: Climate change and mountain water resources: overview and recommendations for research, management and policy, Hydrol. Earth Syst. Sci., 15, 471–504,

https://doi.org/10.5194/hess-15-471-2011, 2011.

Winkler, M., Schellander, H., and Gruber, S.: Snow water equivalents exclusively from snow depths and their temporal changes: the $\Delta$SNOW model, Hydrol. Earth Syst. Sci., 25, 1165–1187, https://doi.org/10.5194/hess-25-1165-2021, 2021.