# Peer review of "Snow depth sensitivity to mean temperature, precipitation, and elevation in the Austrian and Swiss Alps"

_EGUsphere, 2024_

## Referee Comment (RC2)

Review for the paper "Snow depth sensitivity to mean temperature, precipitation, and elevation in the Austrian and Swiss Alps" by Switanek et al

Switanek et al examine the dependence of snow depth (SD) on temperature (T), precipitation (P), and elevation (E) in the Austrian and Swiss Alps. By using historical data from weather stations, they build a statistical model (SnowSens) to estimate seasonal SD based on these predictors. The statistical model is trained with data from 1901-1970/71, then evaluated over 1971/72-2021. The model performance is compared with that of the physics-based model SNOWGRID-CL for a subset of weather stations. Finally, the statistical model is used to estimate SD over the entire domain and some conclusions are drawn on future changes of SD at specified elevation bands.

The authors claim that SnowSens is used to "forecast snow depth" (SD), although SD estimates are produced with contemporaneous observed T and P. The model is, as presented, an emulator of SD driven by P and T, and not a forecasting tool. This and other major concerns listed below diminish the significance of this work, and should be clearly addressed before the paper can be considered for publication:

**Major**

1. SnowSens is not a forecasting tool. SnowSens forecasts could be produced if SD lagged T and P, or T and P were themselves forecast, which is not the case in this study. The authors call "forecasts" what seem to be out-of-sample estimates of SD used to validate their model. Therefore, the authors should give a more clear explanation of how their model should be applied. Is this statistical model expected to outperform more advanced state-of-the-art physics-based models? Or, is it more a diagnostic and analysis tool? Perhaps the authors should emphasize applications such as that discussed in L343-355 and Fig. 12, with estimations of future SD based on projected T and P.

2. The statistical model seems to work best at larger scales (e.g., averages over elevation bands), but it may fail at representing e.g., interannual variability at smaller scales, where processes such as orographic precipitation as well as blowing and sublimation of snow can greatly affect the snowpack. Can the authors comment on this?

3. L322. Related to the previous comment: to provide a comprehensive assessment of the modeled SD "year-to-year variability", it would be beneficial to include results of the anomaly correlation coefficient (ACC) of estimated and observed SD. Given the results in Fig. 10 and the comment in L307-308, ACC for the estimated SD at weather stations may be low. If so, the authors should clearly and explicitly address this shortcoming of their method. I would be curious to know whether (and how) the authors plan to overcome this.

4. Based on Fig. 9, SnowSens tends to underestimate SD more than SnowGrid-CL does, particularly for high SD. This suggests that SnowSens may not work well at estimating

high snow accumulations and more generally in cases of extreme snowfalls. Can the authors comment on whether/how their model could/would handle extreme events?

5. L309-316. Are the values reported in Table 2 for bias corrected SNOWGRID-CL? Please clarify. If not, please provide the bias corrected values as well.

6. How sensitive is the statistical model to the bin size discussed in L194-L204. Is it robust to changing bin sizes?

7. Following on the previous comment, have the authors considered quantifying the uncertainty of their statistical model?

8. I may have missed it, but how the authors extrapolate T, P, and SD in Fig. 5e,f,g,h to create the maps in Fig. 5i,j,k,l beyond the range of observed values? For example, in Fig. 5i for the 0-500 m band, how is the map created for temperature anomalies greater than 3°C or precipitation more than twice the normal values? It seems unlikely that the model will perform well out of the observed range.

9. Table 3 states that the result are statistically significant at $p < 0.05$. What statistical test is used to establish this?

10. L249-259. In describing Figs. 6 and 7, the authors make good points regarding the nature of SD and how those fitted straight lines could be misleading. Another point is that the sample size may be different each year (e.g., there may have been considerably less stations at the start of the recording period compared to present time as is clearly the case for Fig. 6d, making the trend largely uncertain). Can the authors comment on this and provide a measure of uncertainty associated to these straight lines?

**Minor**

1. L1 Delete "incredibly" and "climatic and"

2. L32-39. altitude → elevation

3. L47. "However, these studies suffer ... strong dependence of snow depth on elevation". Please clarify.

4. L50. "This allows us to remove the influence of elevation ...". Please clarify. "Remove" from what? The dependencies established in this study are strongly affected by elevation.

5. L68-71. Please clarify what homogenization means in this context and why one or the other choice is not expected to change the results.

6. L96. Specify what those time series are? Seasonal averages at various years?

7. L121. Delete "in a given month at a given station". Unless I've misunderstood the statement, it refers to the snow depth coverage of the 291 stations for all the Januaries during 1901-2020.

8. L125-127. If so, why not simply use the November-April or the November-May season as in previous studies?

9. L134. precipitaion → precipitation

10. L137. "homogenized stations"? It seems the authors provide a method to homogenize the data, but precipitation and mean temperature are taken over all "available" stations?

11. L141-144. This is not clear. In particular, how is the first of the "two time series" computed? Is the second time series an actual time series or an average value over the training period? And, how is the "first time series" adjusted? Do you mean it is super-imposed to the average temperature computed in (2)?

12. L154. Delete "the similarly"

13. Figure 3 shows correlations between SD and T or P, and their dependence with elevation. Given that T and P are not independent variables, perhaps it would be more illustrative to show partial correlations e.g., between SD and T while controlling for P, and between SD and P while controlling for T. In a way, those partial correlations are related to the partial derivatives over the surface shown in Fig. 5.

14. L165-169. Unlike P and SD, Eq. 4 shows T "anomalies" relative to the climatology over the training period. These anomalies are not normalized. Why are they called "normalized" temperatures? If there is a need to refer to "normalized" T, P and SD with one term, then perhaps use "reduced", or simply normalize the temperature anomalies with a relevant scaling factor common across stations and years.

15. L165-174 Define $\mathbf{T}_{x,t}$, $\mathbf{P}_{x,t}$ and $\mathbf{HS}_{x,t}$. In particular, is $\mathbf{P}_{x,t}$ the *accumulated* or *averaged* precipitation over November-March at station $x$ and year $t$?

16. L177-178. The larger squares are hard to see in the figure. And, what "black lines"?

17. L179-182 "One can observe... two-dimensional plane (not shown)... in the lower-right". This is not clear. What 2D planes?

18. L176-190 This paragraph seems to be a motivation to include an SD dependence not only on T and P, but also on elevation. If so, the explanation could be simplified and made clearer, and previous work explicitly addressing this could be cited, e.g., Moran-Tejeda 2013 [`doi: 10.1002/grl.50463`], Sospedra-Alfonso et al 2015 [`doi:10.1002/2015GL063898`], Scalzitti et al 2016 [`doi:10.1002/2016GL068798`].

19. L207. valus → values

20. L232 and L234. Consider deleting "real-valued" and use only "absolute" value, or "full" value.

21. L241. This is confusing. How are $\overline{\mathbf{HS}}_{\mathrm{MOD}_{1962-1971},x,t}$ and $\overline{\mathbf{HS}}_{\mathrm{OBS}_{1962-1971},x,t}$ in Eq. 8 defined? Do they depend on $t$? And, is the numerator in Eq. 8 missing an $*$?

22. L271. The comparison is for the last 30-year averages relative to averages over a 40-year period. Why not 30 years for consistency? And, are the dots in the figure averages at all available stations? Sampling errors seem to impact more lower than higher elevations.

23. L293. As mentioned above, I wouldn't call this "forecast skill", as these are not actual forecasts. Perhaps refer to it as a measure of model "accuracy" or "performance"?

24. L341-342. In the panels of Fig. 11, the authors give the correlation coefficients computed for the elevation bands and validation period. These correlations are largely driven by the decreasing trend (particularly at lower elevations). Could the authors add the correlations for the detrended time series?

25. L367-368 That SnowSens can "skilfully forecast year-to-year variability of snow depth" seems an overstatement, particularly when ACC at the level station were not provided or discussed.

26. L378 Delete "of the world"

---

## Author Response (AR1)

Reviewer comments are in **black**.

Initial author responses during the discussion period are in **blue**. Line numbers correspond to original paper submission.

Additional author responses are in **green**. Line numbers correspond to the revised version of the paper.

**Reviewer 1:**

Switanek et al. provide an analysis of the multivariate dependence of relative snow depth anomalies over the Austrian and Swiss Alps to temperature and precipitation anomalies. Besides showing past trends of relative snow depth trends, they use the estimated sensitivities to predict snow depth and compare it to a degree-day snow model. The multivariate approach is interesting and has a lot of potential for understanding past changes and predicting future changes. However, some major reservations need to be addressed or discussed first. Finally, it is unclear what the paper is mainly about. I tended to follow what was written in the title. But there are also other elements within that need to be linked to the research aims (a lot of trend analysis of relative changes and comparison to a degree-day model).

The paper's structure is somewhat unfamiliar, because it does not follow the standard approach of intro, methods, results, discussion, but instead guides the reader through a research journey with a lot of motivation used, e.g., in the methods description. Personally, I enjoyed reading it. But, a major drawback is that methods are sometimes difficult to find, since they are spread out. Furthermore important elements are missing, the research questions/aims and the discussion. I honestly don't know, if I should recommend a standard paper structure or not, but definitely the missing components need to be added.

The authors would like to thank the reviewer for his time and effort in providing useful feedback concerning our paper. The reviewer has made a comment about the structure of our paper. We would like to be clear that our paper does contain the standard sections mentioned by the reviewer (i.e., intro, methods, results, discussion/conclusion). The last paragraph in our Introduction outlines the primary focus of the paper. The reviewer has questioned: What is the paper mainly about? Our main goal of the paper is to use observational records to show the sensitivity of snow depth to temperature and precipitation anomalies at different elevations. And furthermore, we show that these empirical-statistical relationships are quite robust over longer periods of time, and as a result we can use historical sensitivities to make surprisingly skillful forecasts of "future" snow depth. One could use a physically-based model to investigate these sensitivities, but they might not align with the observational records themselves. Therefore,

we use the observational data itself to inform us and produce a data-driven model to better quantify these sensitivities. While that is the main focus of the paper, we do also provide some additional trend analysis in order to provide the specific context for the data we used in our study. The second additional component of the paper is the comparison of the forecasts from our proposed methodology to an existing model, SNOWGRID-CL. The authors find these additions to be strengths, rather than a distraction, from the paper. However, if it is seen as beneficial to the paper to remove anything relating to the observed historical trends, we could proceed in that direction.

In an effort to improve the structure of the paper, we have removed the content related to trend analysis from the Results section of the paper. This has been moved to section 3.1.1 along with the addition of Appendix A.

Major points

1. I would expect temperature and precipitation to have different effects in the accumulation and ablation phases of the snow cover. But in your model, using seasonal averages, accumulation and ablation are treated together. Did you perform tests for differences in sensitivities between start and end of the snow season?

   This is a good observation of the reviewer. While the authors agree that greater model complexity has the potential to further improve forecasts, that is precisely what we are trying to avoid in this paper. The main objective of the paper is to show how effective a simplified data-driven, empirical-statistical model performs in making forecasts of long-term changes to snow depth. We consider some of these simplifications (e.g., seasonal averages of our predictors, or using a type of localized linear regression model) to be a strength. This allows researchers and other end-users to very easily visualize how different combinations of changes in precipitation and temperature would be projected to translate into changes in snow depth. As we state at L369: "The SnowSens model is not to be seen as a replacement for physically-based models such as the SNOWGRID-CL." We show how large simplifications can still provide very useful and skillful forecasts, most especially concerning long-term trends averaged over elevation bands.

2. One major drawback of your method is the strong need for extrapolation of the sensitivities in "unknown" climatological terrain. In my opinion, the

chosen approach using local linear regression produces unrealistic values, especially at the boundaries and beyond the training domain (Fig 5a-d). Moreover, it smoothes out a lot of local effects (Fig 5 comparing the different columns); this might be a reason why SnowSens does not capture interannual variability. I don't know a simple remedy to this, but at least this needs to be discussed.

Thank you for this comment, and the authors appreciate your skepticism. It is true that we use extrapolation in our methodology. To be clear, the sensitivities, shown in Figure 5, are constructed only using data over the calibration period 1902-1971. This same period is also used to calibrate our SnowSens model. Then, forecasts of snow depth are evaluated over the validation period 1972-2021. Therefore, if the model was systematically producing unrealistic values, then that would adversely affect our skill measures. We do not find this to be the case. As stated in the paper, we find the trends of the band-averaged forecasts to track very well with observations over the 1972-2021 validation period (see Figure 11).

And yes, we have already pointed out (L341) that the SnowSens model does underestimate the observed interannual variability for any given individual station. Perhaps the authors can do a better job stressing in our revised version of the paper the most appropriate application of our proposed methodology. In our revision, we would more strongly recommend that a user of our methodology should not place too much weight on the forecasts for any one station or any one point location, but rather should focus more on band-averaged forecasts. For the paper, we wanted to be transparent about how the skill of the SnowSens model compares to something like the SNOWGRID-CL model. Therefore, we initially show the interannual skill at the station level.

Here would be a good place to discuss the extrapolation that we use in our model. Later, the reviewer has this comment when discussing L210: "Personally, I would not trust the values far beyond (>1degC, 50% prec) what one sees in Fig 5e-h." In Figure 1, seen below in this response to the reviewer, we have plotted the cases which fulfilled these criteria. Figure 1a shows the 95 cases where the average seasonal temperature in the validation period was greater than 1.0degC and less than 50% of normal precipitation. One can see that there is not perfect agreement between the individual forecasts and observations. That would be true for any snow model. Though, the error of the SnowSens forecasts are less than half of the climatological forecasts (indicated by RMSE_SS>0.50). The average of the forecasts and observations over these cases are the same; they are both 33% of normal. Figure 1b increases the sample size by using a

threshold of less than 75% of normal precipitation. This gives us 988 cases. Again, the average forecast error is less than half of climatological forecasts. The average of the forecasts and observations over these cases are 42% and 41%, respectively. So, while we are extrapolating to "unknown" climatological terrain, we find the model is quite capable of performing well in that new terrain, especially when aggregating over a number of cases.

We have added text, reflecting the above content, in the revised version of the paper between L353-363. Also, a new Figure 10 has been added to the paper.

[Figure]

Paper Figure 10: Figure 10a shows the 95 cases where the average seasonal temperature in the 1972-2021 validation period was greater than 1.0degC and less than 50% of normal precipitation. Figure 10b increases the sample size by using a threshold of less than 75% of normal precipitation. This gives us 988 cases. The larger squares are the forecasted and observed averages over these cases. The skill scores, for these two different criteria, are shown in the top right of the subplots.

3. I understand the choice of elevation bands, but in a changing climate context, I could also imagine a lot of potential for statistical methods to learn across elevation, at least what concerns temperature, given its strong dependence with elevation. However, this probably requires going away from anomalies to absolute temperature and snow depth values. Did you test the multivariate dependency also for "raw", ie., absolute values of

temp, precip and HS? Would it work? Also without subdividing by elevation?

We would like to thank the reviewer for bringing up this point. In the Conclusions, we state at L374: "If these sensitivities continue to remain persistent into the future, then this modeling approach can be expected to yield skillful forecasts for the next 50 years." We only used data from 1902-1971 to forecast snow depths for the period 1972-2021. These forecasts were shown to be skillful. As a result, one can logically conclude that the sensitivities over the last 120 years have been reasonably stationary. Given this information, we then propose that these methods could produce skillful forecasts over the next 50 years. In contrast, we are not proposing that the historical sensitivities be applied for the next 200, or 500, or 1000 years. A user should periodically update the sensitivities, in addition to testing their effectiveness in a cross-validated framework, prior to making another long-range forecast. For example, people in the year 2050 should not solely rely on data from 1902-1971 or 1902-2021 to say something about the future of snow depths. They can, and should, also incorporate data over the more recent period 2022-2049.

The reviewer asked about if "raw" or absolute values can effectively be used. In our study, we found that constructing the SnowSens model using absolute values across either, 1) elevation bands or, 2) all of the stations, produced forecasts that performed substantially worse than the normalized version of the model (the forecasts from the absolute model also performed worse than climatology). We can show why normalization is a critically important step when using our proposed methodological approach. First, take a look at Figures 2a-2c here in this response to the reviewer. The bar plots show the distribution of values for absolute temperature, precipitation, and snow depth for the Austrian and Swiss stations between 500-1000 meters. The average station height of the Austrian stations used is 745m, while it is 742m for the Swiss stations. So, they are not much different in elevation between the two regions. One can observe that the Swiss stations are generally warmer and wetter than their Austrian counterparts. At the same time, the Swiss stations have lower seasonal averages of snow depth. Let's take a further subset of these Austrian and Swiss data points over this 500-1000m elevation band. Those observed data points of the subsets of data are shown as the scatter plots in Figures 2d-2e. A Student's t-test shows that the means (for temperature, precipitation, and snow depth) of the subset of Austrian data points (Figure 2d) are all statistically significantly different than the subset of Swiss data points (Figure 2e). Looking closely, we find that while this subset of historical observations in Austria has a greater absolute temperature and

less absolute precipitation than the Swiss subset, the Austrian stations have significantly more absolute snow depth than the Swiss stations. As we decrease temperature and increase precipitation, we should expect snow depth to increase. However, this is exactly the opposite of what the absolute data is telling us. By simply using the absolute data alone here, we get the wrong signal. This is an example of a regional or spatial climatological difference that we can address through normalization. After normalizing the data, we are able to leverage information across a larger region.

We have added text, reflecting the above content, in the revised version of the paper between L192-207. This text is accompanied by the addition of a new Figure 5 in the paper.

[Figure]

Paper Figure 5: Figures 5a-5c are bar plots that show the distribution of absolute temperature, precipitation, and snow depth for the Austrian and Swiss stations between 500-1000 meters over the historical period 1902-1971. The average station height of the Austrian stations is 745m, while it is 742m for the Swiss stations. The percentages of the blue and orange bars in each subplot (5a-5c) will sum to 100%. The bar plots are comprised of 1,755 observed data points for Austria and 558 data points for Switzerland. A subset of these Austrian and

Swiss data points are shown as the scatter plots in Figures 5d and 5e, respectively. The size of the squares reflect the values of absolute snow depth. So the larger the snow depth, the larger the square. The subset of Austrian data points have a greater absolute mean temperature, less absolute mean precipitation, and greater mean absolute snow depth.

Minor points:

- L1: What climatic cycles do you? Maybe better rephrase, since climatic cycle can mean something like the Milankovitch cycles.

  Yes, we can make this clearer. Though, Milankovitch cycles operate between tens of thousands and hundreds of thousands of years. Since our paper focuses on a time horizon on the order of ~50 years, it would be unlikely that a reader would be confusing these two.

  We have added "annual" into our abstract at L1.

- L40: Might be worth mentioning doi:10.1002/joc.8002 who also attempted something similar for snowfall

  We can do that.

  This is added at L40.

- L49-54: this belongs into methods. Please provide here a more conceptual statement how you go beyond the state-of-the-art and what your research questions, aims, or hypotheses (choose one) are.

  We respectfully disagree with the reviewer here. It is standard practice to provide a brief outline of what is done in the Introduction of the paper. We do provide one research aim. See our initial response above. The trend analysis provides relevant context for our work, while the forecast comparison to an existing model provides a necessary level of legitimacy of our newly proposed methodology.

  As we said above, we have moved the trend analysis out of the Results section in an effort to improve the structure of the paper.

- L99: so y_clim,i should not be a time series but a fixed value for every station, right? Maybe state it explicitly. Also your RMSEs are then the average over all stations?

  y_clim at station, i, can be thought of as a single value or an time series array where all values are the same. We will make this more clear. And yes,

as the equations 1 and 2 indicate, the averages are performed over all of the stations.

We attempt to make this clearer at L101.

- Sec 3 is more than just methods, it contains a lot of background information and motivation

See above. It is primarily trend analysis that is the additional component of the paper. We found this to be a valuable addition to provide the necessary context with the specific data that we are using. However, if it would improve the paper to remove this content, we can consider doing this.

See above.

- L126: Not sure I agree that Nov-Mar performance should equal to Nov-May. See also Major point 1.

We respectfully disagree with the reviewer here. One can easily compute Nov-Mar and Nov-May seasonal averages of snow depth. We have done that, and their similarity can be observed in Figure 2b. This does not mean that the April-May average cannot also have its own variability, it is just that the April-May contributions to Nov-May average snow depth are obscured, to a large extent, by the larger contributions from Nov-Mar. Also, keep in mind that we are showing and comparing the similarity of the normalized quantities, and not their absolute quantities. The normalized quantities Nov-Mar (normalized with respect to Nov-Mar, station-by-station) are very strongly related to Nov-May (normalized with respect to Nov-May, station-by-station). Put another way, when the Nov-Mar average snow depth, for a particular station, was about 20% above average (or 120% of normal), then we can expect that the Nov-May will also be quite close to 20% above average. We compute a mean absolute error between the two normalized seasonal averages of 3.0%. So, on average, a Nov-May percentage anomaly will vary about 3.0% above or below a Nov-Mar percentage anomaly. The Nov-Mar anomalies explain 99% of the variance of the Nov-May anomalies.

We changed the sentence starting at L128 to be clear that we are referring to anomalous values of the seasonal averages. In that case, as stated in the paragraph above, the anomalous quantities from the two seasons are extremely close to one another.

- Sec 3.2. is unclear. Please describe better how you performed the interpolation. Eg, "function of the inverse distance"? "adjusted to match"? Also not clear if your interpolation takes into account the effect of

elevation? The five nearest stations might not be equally representative in that regard.

We say at L142 that we use inverse distance weighting. It is true that elevation can influence the absolute values of these meteorological quantities. However, since we use normalized temperature and precipitation anomalies, it doesn't particularly matter to us or our model what the absolute values of these predictors are. That said, if one were to produce and use "better" predictor data along with our methodology, this should only improve our model performance.

We have tried to improve that section. We have changed some of the text between L163-171.

- Related: Why did you not use LAPrec or the gridded HISTALP to extract this information? They use homogenized input, but at least for LAPrec, the spatialization is much more complex and takes topography well into account.

We made a choice of the data to use for our study and to construct our sensitivity maps. While beyond the scope of our paper, it could be useful for a future study to compare the influence that different data sets have on the results.

- L150ff: Seems like research questions to me, not methods.

We are providing local context, in this section, for the methods that are being presented.

- L156: which correlation coefficient (Pearson, Spearman)?

Good point. We use Pearson correlation. We will make that clearer in the revised version of the paper.

We now state in the paper that Pearson correlation coefficients are used.

- Fig 4: Please do not use rainbow scales, since the changing colors introduce artificial visual breaks. Use a continuous scale such as viridis, scico (https://www.fabiocrameri.ch/colourmaps/), or similar. Moreover, figure looks quite overplotted, maybe it could help to sub-divide by elevation bins? Ok, I see this comes as Fig5. So maybe in Fig4 you could focus on a few single stations instead or omit?

Thank you for the good suggestions. We can think about how to improve the visibility of these figures.

We have changed the colormaps of Figures 6 and 7. For Figure 6, we still want to show all of the values in the record. Then, in Figure 7, we break this up by the different elevation bands.

- L209: how did you define "nearest quartile" in 2d?

  Thank you for pointing this out. We will have to make it clearer what we have done there. We use a Euclidean distance measure which essentially equates the distances of a 10% precipitation anomaly with a 0.2decC temperature anomaly. So, a data point that had the coordinates of (0.4decC warmer, 0% of normal precip) with respect to a point of interest, and another data point with coordinates (0.0decC, 20% of normal precip), would be treated as the same distance. We did not find the model to be overly sensitive to providing more or less weight to the temperature or precipitation axes.

  The text between L256-260 have been added to better describe how we define the nearest quartile.

- L210: Why did you not use the actual values for your localized linear regression instead of the bins? In that way, you can maximize the information better, and also include information beyond empty bins (< 50 values). Moreover, in statistics, extrapolating beyond the range of training data is controversial. Personally, I would not trust the values far beyond (>1degC, 50% prec) what one sees in Fig 5e-h. Finally, since you want to get 2d-surfaces, GAMs (generalized additive models) seem like a prime tool to be used (with a 2d tensor product smooth); it would not require to bin your data, and would also work in 3d with elevation as third predictor.

  See above our answer to major point 2. While the forecasts are not perfect, the authors find that the model performs quite well in the climatological region that you propose. With respect to GAMs and a tensor product: As we have said above, our current aim is to show how something quite simple can still perform quite skillfully. However, as we have also pointed out, increasing model complexity has the potential to further improve on the methods proposed here.

- L242 Please explain, why the bias correction is needed.

  Without bias correction, the SNOWGRID-CL model (which is the one we compare ours against) performs about as well as climatology. This is due to the bias of the SNOWGRID-CL model (see Table 2 in the paper). For example, SNOWGRID-CL might track the interannual variability fairly well for a station, but its forecast average might be twice as large as the observed average. Calculating the error on the uncorrected forecasts will show that the model is not skillful, while the skill of the SNOWGRID-CL model dramatically improves with bias correction.

We had already stated in the initial version of the paper, "Prior to any bias correction, the SNOWGRID-CL predictions perform about as well or worse than climatology." This can be found in the revised version at L311.

- Sec 4.1. Why this? Not related to the main paper goal, I guess? Also there are some methodological concerns, and missing descriptions: related to data coverage, usage of linear regression for multiple stations (not recommended, because of their correlation, better to a regional/elevation series first), why the arbitrary split in two periods given the know non-linearity of change (papers by Marty and co.).

At L251, we discuss a couple of caveats to the calculation of the trends. The authors disagree that the beginning and middle of last century are two completely arbitrary starting points.

- L307: What test did you use to assess this significance of skill?

We used bootstrapping to test for statistical significance. We will be sure to add that into the revised version of the paper.

We have added this. See L322 and L346.

- Fig10 a) and b) scales do not match but should? a) has -0.4 to 0.4 and b) has -0.2 to 0.6

There is one station that was cut off from Figure 10b that corresponds to the red station in Figure 10a. We did this simply to improve the visibility of Figure 10b.

We changed the look of this figure (now Figure 11) to show all of the data points.

- L341: Does this also hold for the single series? Would be interesting to see some single stations time series and not only regional averages.

We need to make more clear where and when our proposed model is most appropriate. For transparency, we compare the year-to-year forecasts, at the station level, to those of SNOWGRID-CL for the Austrian region. However, we propose a user exercise caution in interpreting the forecasts of any one station or point. See above. Rather, we recommend interpreting the results over band-averages.

Yes, there is skill for nearly all of the individual stations, which can be seen in Figures 11a and 11b. You can see all of the modeled versus observed anomalous seasonal snow depths in the validation period in the new Figure 9. A scatter plot allows us to plot all of the modeled versus observed values, instead of just one or two time series from a station or two.

- L350: Very interesting application of your method. However, 3.2degC is beyond your training range for that elevation range, so the accuracy is highly questionable. Especially, since your numbers are very different compared to previous studies (a comparison with existing literature would be very useful, there are a lot of studies using regional climate models, or snow models forced with climate models).

  If you look closely at Figure 5i, it is around temperatures above 3.5degC and below normal precipitation that the model predicts zero precipitation for the elevation band 0-500 meters. While it is true that these criteria are beyond the training range of the data, we find that the model performs quite well in these cases in the validation period. There are 32 instances that fulfill these criteria in the period 1972-2021. As indicated by Figure 5i, the predicted values for these 32 cases is always 0% of normal. The observed values for these 32 cases range between 0%-25% of normal, with a mean of 8% of normal. This translates to an RMSE_SS is equal to 0.89, which means that the error associated with the model is nine times less than climatology. So, while a number of the observed values in these cases are not exactly zero, they are quite close to it.

  While we did include the new content regarding new climatological terrain (as stated above), we chose not to include this example in the revised version of the paper.

- Discussion of results missing.

  We need to make more clear where and when our proposed model is most appropriate. Thank you for this point. We will see where we could expand on our discussion.

  We made some changes to the paragraph between L402-414 in an attempt to clarify what we see as the most appropriate application of our proposed model.

**Reviewer 2:**

Switanek et al examine the dependence of snow depth (SD) on temperature (T), precipitation (P), and elevation (E) in the Austrian and Swiss Alps. By using historical data from weather stations, they build a statistical model (SnowSens) to estimate seasonal SD based on these predictors. The statistical model is trained with data from 1901-1970/71, then evaluated over 1971/72-2021. The model performance is compared with that of the physics-based model SNOWGRID-CL for a subset of weather stations. Finally, the statistical model is used to estimate SD over the entire domain and some conclusions are drawn on future changes of SD at specified elevation bands.

The authors claim that SnowSens is used to "forecast snow depth" (SD), although SD estimates are produced with contemporaneous observed T and P. The model is, as presented, an emulator of SD driven by P and T, and not a forecasting tool. This and other major concerns listed below diminish the significance of this work, and should be clearly addressed before the paper can be considered for publication:

The authors would like to thank the reviewer for their time and effort in providing useful feedback concerning our paper. The reviewer has made a valid point about "forecasting." The authors will change the terminology used in the paper to reflect how the performance of the model is being evaluated.

We have changed "forecast" to "predict" throughout the paper. We have also added the paragraph between L278-288. This discusses how we are making the predictions in our paper, but how we envision forecasts can be made using a range of future projections of temperature and precipitation.

Major

1. SnowSens is not a forecasting tool. SnowSens forecasts could be produced if SD lagged T and P, or T and P were themselves forecast, which is not the case in this study. The authors call "forecasts" what seem to be out-of-sample estimates of SD used to validate their model. Therefore, the authors should give a more clear explanation of how their model should be applied. Is this statistical model expected to outperform more advanced state-of-the-art physics-based models? Or, is it more a diagnostic and analysis tool? Perhaps the authors should emphasize applications such as that discussed in L343-355 and Fig. 12, with estimations of future SD based on projected T and P.

Thank you for this comment. The authors agree that we need to do a better job in explaining how our model should be applied. The main objective of the paper is to show how effective a simplified data-driven, empirical-statistical model performs in making "forecasts" of long-term changes to snow depth. This allows

researchers and other end-users to very easily visualize how different combinations of changes in precipitation and temperature would be projected to translate into changes in snow depth. As we state at L369: "The SnowSens model is not to be seen as a replacement for physically-based models such as the SNOWGRID-CL." We show how large simplifications can still provide very useful and skillful forecasts, most especially concerning long-term trends averaged over elevation bands or new climatological terrain (as Reviewer 1 referred to it).

We do agree that the model values used for validation in the paper are not exactly "forecasts." This is because, as the reviewer has pointed out, we use seasonal temperature (T) and precipitation (P) to say something about contemporaneous or concurrent snow depth (SD) anomalies. However, we are very much proposing that the model and/or the sensitivity plots be used to make actual forecasts of SD given projected future ranges of T and P. We have used a perfect prognosis approach to quantify the uncertainty of one part of the modeling chain as it concerns seasonal snow depth. We have asked the question: Given specific known values of average seasonal temperature and precipitation at the various stations in the region, how accurately can we "forecast" the values of SD? We have answered this in the paper, and our associated skill measures tell us this. As with any model, though, there can be uncertainty that is added anywhere along the modeling chain. If observed future values of T and P differ from what were forecast by CMIP6 for example, then this adds to the uncertainty and will ultimately degrade the quality of the forecasts of any snow model that is used. The whole point of the validation of our model is to give users the confidence to apply it in a true forecasting framework. So, consider an example where we know the average CMIP6 seasonal forecasts of T and P over a period of time such as 2031-2070. Given those conditions, with those specific forecasts of T and P, we can make an actual forecast of SD over that same period of time. We will clarify these points in the revised version of the paper. As a comparison, the SNOWGRID-CL model is also being run with concurrent data. The SNOWGRID-CL model also makes "forecasts" or estimates of SD, for a particular day, given T and P from the same day.

To further illustrate our approach and applicability, consider an illustrative example where some climate model can make forecasts out for the next six months. One might want to observe whether that model is capable of capturing, in a forecast framework, the precipitation patterns associated with ENSO. Initially, the model developers might run the model with reanalysis data in order to observe how well the "known" set of conditions can be used as model input to simulate a "known" set of precipitation observations. If, over many cases, the precipitation patterns do not align very well with ENSO patterns, then perhaps that model should not be trusted in a new and unknown case. On the other hand, if the model is found to perform well, then the model can be useful to provide a

set of actual precipitation forecasts which have been conditioned on a forecast of ENSO (e.g., the NINO3.4 region will be 1.5°C above average over the next 3 months). Our approach is similar. We show, over a set of "known" cases, that the model is quite capable of quantifying how different T and P anomalies translate to anomalies of SD. Now, with a skillful model one can make actual forecasts of SD given different a range of future values of T and P. We will make this clearer in the revised version of the paper.

See our comment in green above.

2. The statistical model seems to work best at larger scales (e.g., averages over elevation bands), but it may fail at representing e.g., interannual variability at smaller scales, where processes such as orographic precipitation as well as blowing and sublimation of snow can greatly affect the snowpack. Can the authors comment on this?

As stated in the paper and above in this response, "The SnowSens model is not to be seen as a replacement for physically-based models such as the SNOWGRID-CL." More advanced state-of-the-art physics-based models have their place, and we are not trying to replace them. While the authors agree that greater model complexity has the potential to further improve forecasts, that is precisely what we are trying to avoid in this paper. We consider some of the simplifications that we use (e.g., seasonal averages of our predictors, or using a type of localized linear regression model) to be a strength. We have already pointed out (L341) that the SnowSens model does underestimate the observed interannual variability for any given individual station. The authors will do a better job stressing in our revised version of the paper the most appropriate application of our proposed methodology. In our revision, we would more strongly recommend that a user of our methodology should not place too much weight on the forecasts for any one station or any one point location, but rather should focus more on band-averaged values. For the paper, we wanted to be transparent about how the skill of the SnowSens model compares to something like the SNOWGRID-CL model. Therefore, we initially show the interannual skill at the station level.

We have tried to be as clear as possible in the revised version of the paper about when and where there is skill in the SnowSens model. Related to a comment of the reviewer below, there is definitely skill in the SnowSens model at the interannual time scale, or when using seasonal values. While the model is clearly skillful at the interannual and station scale, we still find that the bias-corrected SNOWGRID-CL model is more skillful at this scale. This is clearly stated in the

paper. New text discussing the interannual (or year-to-year) variability can be found between L342-352, along with the new Figure 9.

3. L322. Related to the previous comment: to provide a comprehensive assessment of the modeled SD "year-to-year variability", it would be beneficial to include results of the anomaly correlation coefficient (ACC) of estimated and observed SD. Given the results in Fig. 10 and the comment in L307-308, ACC for the estimated SD at weather stations may be low. If so, the authors should clearly and explicitly address this shortcoming of their method. I would be curious to know whether (and how) the authors plan to overcome this.

We thank the reviewer for this comment. As stated above, our methodology is not designed to better capture the year-to-year variability at individual stations in comparison to a more state-of-the-art physics-based model. Given what we have already said about the applicability of our model, we do not advise placing too much weight on RMSE_SS values or anomaly correlation coefficients at individual stations or point locations. We typically avoided using something like ACC (with the exception of Figure 11) because it is not well suited in evaluating skill over trending time series. Additionally, ACC does not show or reflect whether any biases are present within a model with respect to observed time series. In contrast, a skill score such as RMSE_SS evaluates how well modeled values match the observed values (a close match will better minimize the errors between the model and observations), which includes information related to the observed trend, along with whether bias exists in either the mean or the variance. In Figure 1, which can bee seen below in this document, there are a set of 5 synthetic "observed" values which are all below normal (i.e., solid black line). The values of Model 1 (dashed red line) have a high ACC, but have large error residuals from the observations because it is not capturing the systematic mean change, which could result from an underlying trend. Model 2 (dashed-dotted green line) has the highest ACC, but is dramatically underestimating the observed variability. Model 3 (dotted orange line) has the lowest ACC, but the highest RMSE_SS. This tells us that, on average, the squared differences between observations and Model 3 are closer to one another than between observations and Model 1 or Model 2.

[Figure]

Response Figure 1: Four synthetic time series are plotted in order to illustrate how effectively different skill metrics evaluate the performance of a model.

We discuss the year-to-year skill of the SnowSens model both above and below in this document. We find that the SnowSens model is skillful in predicting year-to-year snow depths at the station level. However, we list in the paragraph above (in this document) the benefits to using the RMSE skill score instead of anomaly correlations.

4. Based on Fig. 9, SnowSens tends to underestimate SD more than SnowGrid-CL does, particularly for high SD. This suggests that SnowSens may not work well at estimating high snow accumulations and more generally in cases of extreme snowfalls. Can the authors comment on whether/how their model could/would handle extreme events?

This point is also concerning year-to-year variability. See our comments above to the last two points. The SnowSens model is not making any attempt to forecast extremes, and that is outside the scope of our current study. We are proposing that the SnowSens model be used to make forecasts of expected changes to band-averaged seasonal SD given future projections of T and P. As stated above, we will make this clearer in our revised version of the paper.

Again, see the content from our prior responses to the reviewer.

5. L309-316. Are the values reported in Table 2 for bias corrected SNOWGRID-CL? Please clarify. If not, please provide the bias corrected values as well.

Thank you for this comment. We will make this clearer in the revised version of the paper. The first skill metrics listed for SNOWGRID-CL (which is the first row in Table 2, and it is labeled: SNOWGRID-CL RMSE SS (HS)) are for the absolute or raw modeled values. This tells us that the absolute values of the SNOWGRID-CL model are not particularly skillful, or much closer to observations than climatology. This is due to the fact that the SNOWGRID-CL model contains substantial mean biases. After correcting for these systematic mean biases, the skill can be seen to improve (i.e., this is the third row in Table 2, which is labeled: SNOWGRID-CL RMSE SS (HS$_*$)).

The skill of the SNOWGRID-CL model, using both the absolute values and the bias-corrected values, are included in the paper and in Table 2.

6. How sensitive is the statistical model to the bin size discussed in L194-L204. Is it robust to changing bin sizes?

While the performance of the SnowSens model can change as a function of bin sizes, we do not observe the performance of the model to be overly sensitive to what we consider reasonable choices of bin sizes, given the ranges of the data and the sample sizes. When using the bin sizes from the paper (window with sizes, 0.8degC T and 40% P), the RMSE_SS across all modeled versus observed values is 0.26 (row 4 in Table 2) and RMSE_SS is 0.19 for its ability to capture the SD changes from one period of time to another for all of the stations (row 7 in Table 2). If we use a window size of 1.0degC T and 50% P, then these skill scores are 0.26 and 0.20, respectively. And, if we use a window size of 0.6degC T and 30% P, then these skill scores are 0.26 and 0.17, respectively. While we present a working and skillful model, a user may deem it appropriate in their application to use a different bin size than what we have put forward in the paper.

We added some text regarding the bin size in the paper at L243: "We did experiment using different bin sizes, though we found that the choice of bin size does not strongly affect model performance (please refer to our comments in the paper discussion at https://doi.org/10.5194/egusphere-2024-1172-AC2)."

7. Following on the previous comment, have the authors considered quantifying the uncertainty of their statistical model?

We thank the reviewer for this question. While we appreciate the motivation to quantify all of the uncertainty associated with our model, this is beyond the scope of our study. We have presented a methodology with which we have evaluated the performance of modeled values with respect to observed values. The skill scores that we have listed in the paper do provide a quantification relating to how "certain," or how much variability, there exists in the observations about the modeled values. However, future work can focus on better quantifying the uncertainty of the model as a function of different elevations, T anomalies, and P anomalies.

8. I may have missed it, but how the authors extrapolate T, P, and SD in Fig. 5e,f,g,h to create the maps in Fig. 5i,j,k,l beyond the range of observed values? For example, in Fig. 5i for the 0-500 m band, how is the map created for temperature anomalies greater than 3° C or precipitation more than twice the normal values? It seems unlikely that the model will perform well out of the observed range.

Here, we include some of our responses to Reviewer 1, who also raised a couple of questions about extrapolating to new values. We are planning to integrate much of the following content into the revised version of the paper.

It is true that we use extrapolation in our model. The reviewer has this comment when discussing L210: "Personally, I would not trust the values far beyond (>1degC, 50% prec) what one sees in Fig 5e-h." In Figure 2, seen below in this response to the reviewer, we have plotted the cases which fulfilled these criteria. Figure 2a shows the 95 cases where the average seasonal temperature in the validation period was greater than 1.0degC and less than 50% of normal precipitation. One can see that there is not perfect agreement between the individual forecasts and observations. That would be true for any snow model. Though, the error of the SnowSens forecasts are less than half of the climatological forecasts (indicated by RMSE_SS>0.50). The average of the forecasts and observations over these cases are the same; they are both 33% of normal. Figure 2b increases the sample size by using a threshold of less than 75% of normal precipitation. This gives us 988 cases. Again, the average forecast error is less than half of climatological forecasts. The average of the forecasts and observations over these cases are 42% and 41%, respectively. So, while we are extrapolating to "unknown" climatological terrain, we find the model is quite capable of performing well in that new terrain, especially when aggregating over a number of cases.

And here is another example of extrapolating to large temperature anomalies. If you look closely at Figure 5i (in the paper), it is around temperatures above 3.5degC and below normal precipitation that the model predicts zero precipitation for the elevation band 0-500 meters. While it is true that these criteria are beyond the training range of the data, we find that the model

performs quite well in these cases in the validation period. There are 32 instances that fulfill these criteria in the period 1972-2021. As indicated by Figure 5i, the predicted values for these 32 cases is always 0% of normal. The observed values for these 32 cases range between 0%-25% of normal, with a mean of 8% of normal. This translates to an RMSE_SS is equal to 0.89, which means that the error associated with the model is nine times less than climatology. So, while a number of the observed values in these cases are not exactly zero, they are quite close to it.

[Figure]

Paper Figure 10: Figure 10a shows the 95 cases where the average seasonal temperature in the 1972-2021 validation period was greater than 1.0degC and less than 50% of normal precipitation. Figure 10b increases the sample size by using a threshold of less than 75% of normal precipitation. This gives us 988 cases. The larger squares are the "forecasted" or simulated and observed averages over these cases. The skill scores, for these two different criteria, are shown in the top right of the subplots.

We have added text, reflecting the above content, in the revised version of the paper between L353-363. A new Figure 10 has also been added to the paper.

9. Table 3 states that the result are statistically significant at p< 0.05. What statistical test is used to establish this?

We used bootstrapping to test for statistical significance. We will be sure to add that description into the revised version of the paper.

We have added this. See L322 and L346.

10. L249-259. In describing Figs. 6 and 7, the authors make good points regarding the nature of SD and how those fitted straight lines could be

misleading. Another point is that the sample size may be different each year (e.g., there may have been considerably less stations at the start of the recording period compared to present time as is clearly the case for Fig. 6d, making the trend largely uncertain). Can the authors comment on this and provide a measure of uncertainty associated to these straight lines?

Thank you for this comment. We have already accounted for the fact that each year can have a different number of measurements. This is done by fitting a least-squares regression line to all of the scatter points in Figures 6 and 7 in the paper (independently for each elevation band), instead of a least-squares fit to the regionally-averaged SD. Yes, if a particular year only had five measurements and another year had 100, then getting regional averages would apply equal weighting to the both of these years even though they have dramatically different data coverage. In Figure 3, below in this document, we have provided a simplified example using synthetic data. In this example there are two individual data points for times 1 and 2, while there are five individual data points for times 3 and 4. One can fit a regression line to the individual blue points, and this is shown as the blue line. This is what we have done in the paper. In contrast, one could average the values at each time and then fit the regression to those time-averaged values (shown as the red line, this could be thought of as the average of all of the available stations for a given season). Because we are not using a very large number of points in this example, the blue and red lines are not all that different. We are only illustrating that they are different, and it does matter what you are using to fit your regression.

The reviewers comment on the uncertainty of the trend is a good one. In the revised version of the paper, we will be sure to add confidence intervals to the plotted trend lines.

We have added some additional description and confidence intervals concerning the plotted trends. See the caption for what is now the new Figure 3 in the paper.

[Figure]

Response Figure 2: Synthetic data which has two individual data points (blue circles) for times 1 and 2, and five individual data points (also blue circles) for times 3 and 4. The red squares are averages of the blue circles at each time step. The blue regression line is fitted to the individual blue points, while the red regression line is fitted to the red points.

Minor

1. L1 Delete "incredibly" and "climatic and"

We can remove "incredibly." However, we see snow depth as something that not only results from a particular climatic regime, but it also influences the climate. Consider the influence of albedo on the energy present on the ground or in the atmosphere, and how that changes as a function of SD.

We have removed "incredibly."

2. L32-39. altitude → elevation

Thank you. We can be more consistent with the wording.

We now use "elevation" throughout the paper.

3. L47. "However, these studies suffer ... strong dependence of snow depth on elevation". Please clarify.

We are primarily referring to some prior work conflating the absolute changes in snow depth and the fact that higher elevations climatologically have greater depths of snow to begin with. At L36, we write: "However, this altitudinal dependence of the snow depth trends is conflated to some degree with the fact that stations at higher elevations also typically receive more snow."

See our response in the paragraph above, and we have changed the wording here and the new content starting at L46 reads: "However, these studies also experience to some extent a conflation between snow depth quantities and elevation. Furthermore, the statistical relationships shown are often correlations, which do not capture how much snow depth would change, for example, as a function of air temperature."

4. L50. "This allows us to remove the influence of elevation ...". Please clarify. "Remove" from what? The dependencies established in this study are strongly affected by elevation.

We remove the influence of elevation on the absolute values of SD. Even after removing this influence, there is still an influence of elevation in a relative sense. We will clarify this point in the revised version of the paper.

We have changed our text to now read: "This step removes regional and elevation-dependent climatological differences, thereby allowing us to better quantify anomalous or relative changes."

5. L68-71. Please clarify what homogenization means in this context and why one or the other choice is not expected to change the results.

Homogenization in this context is referring to the possible shifting up or down different parts of the snow depth time series at certain stations. This can happen if they were found to experience a noticeable break (or shift) in the time series due to something like moving the station up or down in elevation. We state at L69, "Through personal communication, the authors of a recent homogenization study in the Alps (i.e., Resch et al. (2022)) have indicated that there are not any systematic changes in snow depth one way or the other as a result of the homogenization procedure (Marcolini et al., 2019; Buchmann et al., 2022)." For that reason, we do not expect the results in our paper to differ all that much from some potential future study that chooses to use homogenized data.

6. L96. Specify what those time series are? Seasonal averages at various years?

That is a good point. We will make this more clear that we are using time series of seasonal averages.

We have added this in. See L96.

7. L121. Delete "in a given month at a given station". Unless I've misunderstood the statement, it refers to the snow depth coverage of the 291 stations for all the Januaries during 1901-2020.

We will remove that text.

We have done that.

8. L125-127. If so, why not simply use the November-April or the November-May season as in previous studies?

This is explained in the paper at L116, "During warmer months, and especially with stations at lower elevations, an observable amount of precipitation will not always translate to a measured snow depth. This would result in trying to fit a predictor time series (i.e., precipitation), which does vary, with a predictand time series that does not (i.e., snow depth). Therefore, we would like to minimize the number of cases where there is zero measured snow depth." And since many more stations can have zero recorded snow depth in the months of April and May, we chose not to include those months in our model fit. That way, we can have consistency in the lengths of our seasons for both the predictors, T and P, and our predictand, SD.

9. L134. precipitaion → precipitation

We will fix that.

We have done that.

10. L137. "homogenized stations"? It seems the authors provide a method to homogenize the data, but precipitation and mean temperature are taken over all "available" stations?

We use temperature and precipitation station data that has already been homogenized by the data provider. We do not apply any homogenization ourselves in the paper.

11. L141-144. This is not clear. In particular, how is the first of the "two time series" computed? Is the second time series an actual time series or an average value over the training period? And, how is the "first time series" adjusted? Do you mean it is super-imposed to the average temperature computed in (2)?

We will rewrite these sentences in an effort to make it more clear to the reader.

We have tried to improve that section. See L161-172.

12. L154. Delete "the similarly"

We will fix the wording there.

We have done that.

13. Figure 3 shows correlations between SD and T or P, and their dependence with elevation. Given that T and P are not independent variables, perhaps it would be more illustrative to show partial correlations e.g., between SD and T while controlling for P, and between SD and P while controlling for T. In a way, those partial correlations are related to the partial derivatives over the surface shown in Fig. 5.

While our model is not very complex, we do want to layer some of the methodological concepts, piece by piece. We first transform the data into anomalies with respect to average T, P, or SD. Then, we show, like others have before, the correlations between SD and T along with SD and P. This initially presents "forecasting" SD as a one-dimensional problem as either a function of T or P. As you say, using either T or P, independently, only gives partial information about what we can expect with SD. That is why we then introduce Figure 4, and show the problem as two-dimensional. Then, in Figure 5, we break up the data into the elevation bands, and now we present SD forecasting as a three-dimensional problem.

14. L165-169. Unlike P and SD, Eq. 4 shows T "anomalies" relative to the climatology over the training period. These anomalies are not normalized. Why are they called "normalized" temperatures? If there is a need to refer to "normalized" T, P and SD with one term, then perhaps use "reduced", or simply normalize the temperature anomalies with a relevant scaling factor common across stations and years.

In order to avoid confusion, the authors changed our terminology in the revised version of the paper. We will present these values simply as anomalies, either as percentages or degrees C from normal.

We have done that.

15. L165-174 Define T x,t , P x,t and HS x,t . In particular, is P x,t the accumulated or averaged precipitation over November-March at station x and year t?

Yes, Px,t is the accumulated precipitation over November-March at station x and year t. At L137, we state, "First, we obtain November-March sums of precipitation and averages of mean temperatures at all of homogenized stations over the years 1901/02-2020/21." So, we use sums or accumulations of precipitation. Though the results would not change if one were to instead use average monthly precipitation. This would simply scale the precipitation accumulations by a common factor.

16. L177-178. The larger squares are hard to see in the figure. And, what "black lines"?

The larger squares are made using black lines. We will work to make this figure easier to see and interpret.

See the revised Figure 6.

17. L179-182 "One can observe... two-dimensional plane (not shown)... in the lower-right". This is not clear. What 2D planes?

Consider the example that we have outlined. We have two predictors, T and P, which are being used to fit some model that can be used to "forecast" SD. With a multiple linear regression fit, then the value of SD depends on both T and P. When one predictor is used, the fit is a line. When two predictors are used, the fit is a plane or a surface. That is what we are referring to. We will improve our description of this in the revised version of the paper.

We have removed that wording, and have now added this sentence at L221: "As one would expect, the average snow depth anomalies increase as the temperature anomaly decreases and the precipitation anomaly increases."

18. L176-190 This paragraph seems to be a motivation to include an SD dependence not only on T and P, but also on elevation. If so, the explanation could be simplified and made clearer, and previous work explicitly addressing this could be cited, e.g., Moran-Tejeda 2013 [doi:10.1002/grl.50463], Sospedra-Alfonso et al 2015 [doi:10.1002/2015GL063898], Scalzitti et al 2016 [doi:10.1002/2016GL068798].

We thank the reviewer in providing us with some relevant citations. We will look if we can make the explanation clearer.

We have added the citations to the Introduction. Also, see L219-233.

19. L207. valus → values

We will change that.

We have done that.

20. L232 and L234. Consider deleting "real-valued" and use only "absolute" value, or "full" value.

Thank you. We can consider that suggestion.

We have removed "real-valued."

21. L241. This is confusing. How are HS MOD 1962−1971 ,x,t and HS OBS 1962−1971 ,x,t in Eq. 8 defined? Do they depend on t? And, is the numerator in Eq. 8 missing an ∗?

Thank you for bringing this to our attention. We will work to make this clearer.

We did remove the "t" subscript. Thank you for catching that. However, the rest of the equation is correct.

22. L271. The comparison is for the last 30-year averages relative to averages over a 40-year period. Why not 30 years for consistency? And, are the dots in the figure averages at all available stations? Sampling errors seem to impact more lower than higher elevations.

We chose to use a longer prior period (the 40-year period) in order to increase the robustness of the measured changes. We could use the most recent 30-year period compared to the 30-year period before, though this will be more subject to sampling variability than using a longer prior period. Though, if it is seen to make more sense to use 30 years for both periods, we can proceed in that direction.

We have chosen to stick with a longer prior period because of what we have said above concerning the robustness of that approach.

Yes, the dots are the percentage anomalies at all available stations. And no, these are not sampling errors that we observe at the lower elevations. This is instead an indication that the snow depth at lower elevations exhibits both greater variability and skewness than the snow depth at higher elevations. It also reflects the zero-bounded nature of something like snow depth or precipitation. The average seasonal snow depth for a station below 500 meters might be something quite close to zero, like 2cm. Some season might be 1cm (or 50% of normal), while another season might be 10cm (or 500% of normal). One can observe the same phenomenon for summer precipitation across the state of California. The average daily precipitation amounts are very small, where the daily averages in July are typically less than 1mm, for example. Then, in the rare events where something like 10mm of rain falls at a station, that would be categorized as an event that is greater than 1000% of normal.

23. L293. As mentioned above, I wouldn't call this "forecast skill", as these are not actual forecasts. Perhaps refer to it as a measure of model "accuracy" or "performance"?

We have discussed this above.

See our responses above.

24. L341-342. In the panels of Fig. 11, the authors give the correlation coefficients computed for the elevation bands and validation period. These correlations are largely driven by the decreasing trend (particularly at lower elevations). Could the authors add the correlations for the detrended time series?

We respectfully disagree with the reviewer. The trends are not responsible for the level of correlations that we report in Figure 11. The correlations that we reported in Figure 11 between the modeled and observed band averages in the validation period are 0.89, 0.84, 0.81, and 0.75, respectively for the four elevation bands. If we detrend both the modeled and observed time series for each of the bands, then we obtain correlations of 0.89, 0.83, 0.79, and 0.74. They are very close to the correlations which contain trends. We can write this in the revised version of the paper.

We provide the detrended correlations in the caption for what is now Figure 12.

25. L367-368 That SnowSens can "skillfully forecast year-to-year variability of snow depth" seems an overstatement, particularly when ACC at the level station were not provided or discussed.

We respectfully disagree with the reviewer. We do not consider that particular text to be an overstatement. The SnowSens model performs statistically significantly better than climatology, which is the baseline measure of skill. The ACC between the SnowSens modeled anomalies versus the observed anomalies is 0.593. This is the result of taking the ACC of all of the modeled vs observed pairings, or the yellow points, in Figure 9 from the paper. We have simplified the viewing of these SnowSens modeled vs observed values, and that is provided as Figure 4 in this document (see below). The number of modeled/observed pairings in the validation period is 10,985. Given that the stations are not independent of one another, the effective sample size would be smaller than 11,064. While the effective sample size might be on the order of three times smaller, we can go to an extreme to clarify our point. Let's instead reduce our sample size by a factor of 100. An ACC of 0.593, with a sample size of 110, has p-value of <1e-13. So, this can be interpreted to mean that there is less than 1 in a trillion chance, on average, that one could achieve the level of skill exhibited by our model, using randomly generated "forecasts." We also produced 10,000 randomly generated simulations, and the maximum ACC value we found in those 10,000 simulations was 0.037. We would argue that it is not in any way an overstatement to say that the SnowSens model has skill when evaluating the performance of year-to-year values at the station level.

As we just pointed out above, Figure 9 shows all of the pairings of modeled values vs observations for the validation period across the four elevation bands. Figure 10 and Table 2 also provide the skill at the station level. We discussed above why we used RMSE_SS instead of ACC.

See our response above. We have included this content above between L342-352, along with the new Figure 9.

[Figure]

Paper Figure 9: All of the modeled values and observed values in the validation period at the station level. There are 11,064 observed cases. If we had 100% data coverage for the observations over the last 50 seasons, then there would be 14,550 cases (50*291). The Pearson correlation coefficient between the modeled and observed values in this figure is 0.61. If one computed the correlation, station-by-station, then the average correlation across the stations is 0.66, with a minimum of 0.29 and a maximum of 0.87.

26. L378 Delete "of the world"

We can do that.

We have done that.

---

## Referee Report (RR1)

Second Review for the paper "Snow depth sensitivity to mean temperature, precipitation, and elevation in the Austrian and Swiss Alps" by Switanek et al

While I appreciate the authors' effort to address the issues raised in the previous review, I found some answers unsatisfactory. The following comments still require, in my opinion, the authors' attention.

**Major**

1. The authors state: "We have changed "forecast" to "predict" throughout the paper. We have also added the paragraph between L278-288. This discusses how we are making the predictions in our paper, but how we envision forecasts can be made using a range of future projections of temperature and precipitation."

   I find the explanation in L278-288 cumbersome and should be simplified. As pointed out in the first review, the authors provide estimates of snow depth anomalies based on anomalies of contemporaneous temperature and precipitation. This makes SnowSens a statistical model or emulator, but not a prediction tool. As per the AMS Glossary of Meteorology: `https://glossarytest.ametsoc.net/wiki/Predictability`, predictability is "the extent to which future states of a system may be predicted based on knowledge of current and past states of the system". I do not deny that the emulator could be driven by future meteorology (obtained from, say, a climate model) to estimate future snow depth, but this is, in my opinion, secondary to this paper and doesn't make SnowSens a forecasting or prediction tool. While the authors should point out those potential applications in the paper, I recommend, for accuracy and clarity, that the authors frame their work as the development of a statistical tool to *estimate* snow depth from meteorological and elevation data.

2. Regarding the authors reply to my previous comment: "The statistical model seems to work best at larger scales (...), but it may fail at representing e.g., interannual variability at smaller scales, where processes such as orographic precipitation as well as blowing and sublimation of snow can greatly affect the snowpack."

   The authors state: L343-344 "... we want to be clear that the SnowSens model still exhibits substantial skill for the year-to-year seasonal predictions at the station scale", and then refer to Fig. 9 to support their claim showing "all seasonal SnowSens predictions versus observations over the validation period using all of the stations in the study domain" pooled in a scatter plot. I contend that Fig. 9 does not support the claim, as it doesn't look at the temporal skill of individual stations, but all times and stations combined. For instance, what would be the result of showing the anomaly correlation coefficient of the (perhaps detrended) snow depth estimates over the validation period at each station individually? I do not suggest to do this, since the emphasis of the paper is on larger scales, but the authors should avoid such statements and, at the very least, recognize in the paper the challenges and limitations of using their tool at station/regional scales, where blowing and sublimation

of snow can greatly affect the snowpack, and temperature and precipitation alone may not suffice as predictors (e.g., Sexstone et al 2018 `https://doi.org/10.1002/2017WR021172`).

3. Regarding the question about extrapolation of $T$, $P$, and $SD$ in (old) Fig. 5e,f,g,h to create the maps in (old) Fig. 5i,j,k,l beyond the range of observed values. In their response, the authors refer to L353-363 and Fig. 10 in the new MS. I am not convinced that this figure shows the "effectiveness of the SnowSens model in its ability to predict in new climatological terrain". Here again the authors pool several stations (and years?) in a scatter plot to support their claim. The authors state L362-363 "while we are extrapolating to "unknown" climatological terrain, we find the model is quite capable of performing well in that new terrain." I'm not convinced that the dispersed cloud of points in Fig. 10 justifies such a strong claim. I think the authors should, at the very least, mention that extrapolation from the trained ranges should be taken with caution.

**Minor**

1. What "annual climatic cycles"? And, consider deleting "annual" as the statement applies to any time scale, e.g., consider instead "snow depth is an important component of the hydrological cycle and the climate".

2. L158: "precipitaion → precipitation". This was mentioned in the previous report (old L134) and wasn't corrected.

3. L208-L218 Define the symbols $T_{x,t}$, $P_{x,t}$, and $HS_{x,t}$. This was pointed out in the previous report (old 165-174) and it wasn't made clear in the MS.

4. Caption to Figure 6 "anomlies → anomalies". Also in L219. Please check text throughout.

5. For the answer to item #22 in the previous review referring to comparisons over periods of different lengths. The authors state: "We chose to use a longer prior period (the 40-year period) in order to increase the robustness of the measured changes...". What robustness? For consistency, it is preferable to compare the statistics over two periods of equal length, particularly when the data have an underlying trend and no statistical significance is provided. If the periods need to be different, please justify.

6. L42-45 The authors state "There have been several prior studies that have linked changes in snow depth, at different elevations, across the Alps to changes in air temperature and precipitation", and then provide several references. Although the references are pertinent, note that they are not all specific to the Alps.

---

## Author Response (AR2)

Reviewer 1:

I would like to thank the authors for replying extensively to all issues that were raised. They added discussion elements to all critical comments, which helps making their study more trustworthy.

We would again like to thank this reviewer for his time and effort in reviewing our paper.

Finally, two more comments:

1. Good choice of color scale; a diverging one makes perfect sense since 100% is equal to normal conditions. Please make sure that the color midpoint (where the scale changes from red to blue) matches the value midpoint (100%, where it changes from negative to positive anomalies). At the moment, it seems that the color midpoint is at ~120% or so.

We have changed these plots again.

2. Figure 13 would need some context with existing literature, and comparison of your sensitivities versus results found in previous literature. 10.5194/tc-11-517-2017 is a good choice, but also 10.1002/joc.4205 or 10.1007/s00382-012-1545-3 and many others.

We have added in a couple of these citations and added this sentence at L390: "These expected changes to snow depth, given different temperature and precipitation scenarios, can be compared to prior studies such as Schmucki et al. (2014) and Marty et al. (2017a).

Reviewer 2:

Second Review for the paper "Snow depth sensitivity to mean temperature, precipitation, and elevation in the Austrian and Swiss Alps" by Switanek et al. While I appreciate the authors' effort to address the issues raised in the previous review, I found some answers unsatisfactory. The following comments still require, in my opinion, the authors' attention.

We would again like to thank this reviewer for his/her time and effort in reviewing our paper.

Major

1. The authors state: "We have changed "forecast" to "predict" throughout the paper. We have also added the paragraph between L278-288. This discusses how

we are making the predictions in our paper, but how we envision forecasts can be made using a range of future projections of temperature and precipitation." I find the explanation in L278-288 cumbersome and should be simplified. As pointed out in the first review, the authors provide estimates of snow depth anomalies based on anomalies of contemporaneous temperature and precipitation. This makes SnowSens a statistical model or emulator, but not a prediction tool. As per the AMS Glossary of Meteorology: https://glossarytest.ametsoc.net/wiki/Predictability, predictability is "the extent to which future states of a system may be predicted based on knowledge of current and past states of the system". I do not deny that the emulator could be driven by future meteorology (obtained from, say, a climate model) to estimate future snow depth, but this is, in my opinion, secondary to this paper and doesn't make SnowSens a forecasting or prediction tool. While the authors should point out those potential applications in the paper, I recommend, for accuracy and clarity, that the authors frame their work as the development of a statistical tool to estimate snow depth from meteorological and elevation data.

We have changed the terminology to "estimates" throughout the paper.

2. Regarding the authors reply to my previous comment: "The statistical model seems to work best at larger scales (...), but it may fail at representing e.g., interannual variability at smaller scales, where processes such as orographic precipitation as well as blowing and sublimation of snow can greatly affect the snowpack." The authors state: L343-344 "... we want to be clear that the SnowSens model still exhibits substantial skill for the year-to-year seasonal predictions at the station scale", and then refer to Fig. 9 to support their claim showing "all seasonal SnowSens predictions versus observations over the validation period using all of the stations in the study domain" pooled in a scatter plot. I contend that Fig. 9 does not support the claim, as it doesn't look at the temporal skill of individual stations, but all times and stations combined. For instance, what would be the result of showing the anomaly correlation coefficient of the (perhaps detrended) snow depth estimates over the validation period at each station individually? I do not suggest to do this, since the emphasis of the paper is on larger scales, but the authors should avoid such statements and, at the very least, recognize in the paper the challenges and limitations of using their tool at station/regional scales, where blowing and sublimation of snow can greatly affect the snowpack, and temperature and precipitation alone may not suffice as predictors (e.g., Sexstone et al 2018 https://doi.org/10.1002/2017WR021172).

This comment makes the claim that "Figure 9 doesn't look at the temporal skill of the individual stations." While Figure 9 doesn't explicitly show the temporal skill of the individual stations, by themselves station by station, Figures 11a and 11b does exactly this. The reviewer keeps bringing up anomaly correlations as one way that we could show this skill station by station. It seems that the reviewer did not find the time to read our response carefully, or understand what we have said, regarding anomaly correlations. We explained in our response to this reviewer why we have chosen to use a skill metric such as the root mean

squared error skill score, and why that, in our case, it is a superior metric to use. This is because, critically, the RMSE skill score accounts for any mean bias that may exist between the modeled and observed values. One can have a time series of modeled values at a particular station that are highly correlated with observations, but the mean of the modeled distribution could be half of what we find in the observed distribution, for example. In cases like that, an anomaly correlation will not suffice in telling us whether or not we have model skill. See Figure 1, at the bottom of this document, for a real example from our study. Figure 1 plots the modeled and observed time series of the station where our model skill is the worst. It has a RMSE skill score of -0.51, while at the same time the anomaly correlation is quite high, at 0.61. Our SnowSens model is capturing the observed variability fairly well in this case, but it has overestimated the expected average decrease in snow depth with respect to what actually occurred. As a result, our skill score in this case was worse than climatology.

Next, we are confused as to why the reviewer believes that our Figure 9 "does not support the claim that the SnowSens model still exhibits substantial skill for the year-to-year seasonal predictions at the station scale." Yes, that figure does pool our entire set of modeled values and plots them against observations. Though, to be clear, each scatter point is a modeled/observed value corresponding to an individual season for an individual station. The scatter points in Figure 9 are comprised of all of the individual time series (modeled and observed) for all of the stations in the validation period. Yes, it is possible in Figure 9 that a station for which the SnowSens model is not skillful can be obscured by the cloud of all of the other stations for which we do have skill. Though, if enough modeled values at individual stations were not skillful in Figure 9, then we would not observe the level of skill that we do. Figure 9 gives us a view of the average skill of our model in predicting season-station anomalies, as measured across our entire study domain and for our validation period 1972-2021. Figure 9 could be thought of as a regionally averaged measure of our interannual skill at the station level. As mentioned above, Figure 9 does not explicitly look at the temporal skill station by station, but Figures 11a and 11b show precisely that. Those are the skills across time in the validation period for every station that we use in our study. Those are more local measures of skill. It seems the reviewer did not see that we have already shown the temporal skill for all of the individual stations. In the paper, we already applied a statistical significance test for the regional skill provided in Figure 9 (L149 in the paper states that we find that this regionally averaged skill is statistically significant with a p-value much less than 0.01, because we generated 10,000 bootstrapped samples and none of them were even close to the skill we observe with our model). We used bootstrapping again to test the statistical significance of the forecast skill at each individual station (Figs. 11a and 11b). We find that there are only a total of 11 stations which either perform worse than climatology or worse than randomly simulated time series (using a p-value < 0.01). Therefore, the SnowSens modeled time series of snow depths from more than 95% of the stations in our study are found to exhibit positive skill and be statistically significantly skillful. Figures 11a and 11b from the paper already

show the temporal skill of the individual stations, and more than 95% of those stations contain statistically significant positive skill.

We have added some relevant statistics to the paper. See L372-376.

3. Regarding the question about extrapolation of T , P , and SD in (old) Fig. 5e,f,g,h to create the maps in (old) Fig. 5i,j,k,l beyond the range of observed values. In their response, the authors refer to L353-363 and Fig. 10 in the new MS. I am not convinced that this figure shows the "effectiveness of the SnowSens model in its ability to predict in new climatological terrain". Here again the authors pool several stations (and years?) in a scatter plot to support their claim. The authors state L362-363 "while we are extrapolating to "unknown" climatological terrain, we find the model is quite capable of performing well in that new terrain." I'm not convinced that the dispersed cloud of points in Fig. 10 justifies such a strong claim. I think the authors should, at the very least, mention that extrapolation from the trained ranges should be taken with caution.

For this point, the reviewer says, "I am not convinced that Figure 10 shows the 'effectiveness of the SnowSens model in its ability to predict in new climatological terrain.'" The reviewer seems focused on the "dispersed cloud of points in Figure 10." However, it also matters where the cloud is situated. Are the values primarily above or below 100% of normal? Our null hypothesis throughout the paper is that a future value can either always be 100% of normal (i.e., climatology), or could be drawn from the distribution of values over the calibration period for the station of interest. Bootstrapping can again be used to assess our statistical significance of these modeled values in Figure 10. In Figure 10, from the paper, the correlation from the 95 points in Fig. 10a is 0.40, and it has a RMSE skill score of 0.62. Similarly, the correlation from the 988 points in Fig. 10b is 0.34, and it has a RMSE skill score of 0.55. Using bootstrapping with the data from the calibration period, we obtain a correlation and RMSE skill score (at $p<0.01$) of 0.24 and -0.01, respectively, in the case of the 95 points of Fig. 10a. Likewise, we obtain a correlation and RMSE skill score (at $p<0.01$) of 0.08 and -0.30, respectively, in the case of the 988 points of Fig. 10b. The RMSE skill scores through randomly generated simulations are not even remotely close to the skill that we find through our model. This is because the SnowSens model is capturing both some of the observed variability, and also the mean shift. Notice where most all of the points are situated in Figs. 10a and 10b. They are almost all below the 100% of normal for both the modeled and observed values. Even if we focus on anomaly correlation alone, which the reviewer refers to as a "dispersed cloud," then the correlations are still statistically significant ($p<0.01$). Perhaps the reviewer has an issue with us using the word "well" as in, "we find the model is quite capable of performing well in that new terrain." We acknowledge that a word such as that has a level of subjectivity. Maybe we can consider changing that phrase to something such as, "we find the model is quite capable of performing skillfully in that new terrain," because that is not debatable. See L365.

We have added some relevant statistics to the paper. See L360-365.

Minor

1. What "annual climatic cycles"? And, consider deleting "annual" as the statement applies to any time scale, e.g., consider instead "snow depth is an important component of the hydrological cycle and the climate".

We have changed this sentence at L1 to, "Snow depth plays an important role in the seasonal climatic and hydrological cycles of alpine regions." This is then better paired with our next sentence which is also referring to seasonal values.

2. L158: "precipitaion → precipitation". This was mentioned in the previous report (old L134) and wasn't corrected.

We have corrected this.

3. L208-L218 Define the symbols $T_{x,t}$, $P_{x,t}$, and $HS_{x,t}$. This was pointed out in the previous report (old 165-174) and it wasn't made clear in the MS.

We have now defined these. See L211-219.

4. Caption to Figure 6 "anomlies → anomalies". Also in L219. Please check text throughout.

We have corrected this.

5. For the answer to item #22 in the previous review referring to comparisons over periods of different lengths. The authors state: "We chose to use a longer prior period (the 40-year period) in order to increase the robustness of the measured changes...". What robustness? For consistency, it is preferable to compare the statistics over two periods of equal length, particularly when the data have an underlying trend and no statistical significance is provided. If the periods need to be different, please justify.

We now use two 30-year periods. See Appendix A and Figure A2.

6. L42-45 The authors state "There have been several prior studies that have linked changes in snow depth, at different elevations, across the Alps to changes in air temperature and precipitation", and then provide several references. Although the references are pertinent, note that they are not all specific to the Alps.

We have just removed "Alps" from that sentence. See L42.

[Figure]

Figure 1: (a) plots the modeled and observed time series, in the validation period, for one station from our study. This individual station has the worst model performance that we found with a RMSE skill score of -0.51 (see Figures 11a and 11b from paper). (b) plots these two time series from (a) as a scatter plot. The anomaly correlation between the two time series is 0.61.

---

## Author Response (AR3)

Dear Matthew Switanek, dear Co-authors,

Thank you for uploading your revised manuscript. I considered all comments and your answers to both reviewers in my assessment. Overall, I find the revisions acceptable in this form, in particular as you explained very clearly where and how the requested information is given (e.g., in which figure).

We would like to thank the editor for her time and her careful assessment of our paper.

At this stage, I ask you just for a short minor revision regarding two points (see below regarding 2 and 3) before acceptance for publication.
Comments from my side regarding the three major concerns of Reviewer 2 and your answers:
1) It makes sense that you changed the terminology to "estimates".

2) I agree with your argumentation and your answer to the major part of this comment. However, I agree with the last point of the comment of the reviewer that you should mention in the manuscript at least that 'blowing and sublimation of snow can greatly affect the snowpack, for that temperature and precipitation alone may not suffice as predictors'. Please add this point in a sentence in the discussion.

We have added in this point in the discussion. See L410.

3) I agree with the author's suggestion to rephrase the sentence to 'we find the model is quite capable of performing skillfully in that new terrain.' However, I would add that results still should be taken with some caution (as this would be the case with any/most (empirical-statistical) models).

We have added an additional sentence at L366.

I am looking forward to your submission of the revised manuscript.

Best regards,
Franziska Koch